# Efficient Training-Free Multi-Token Prediction via Embedding-Space Probing

**Raghavv Goel** [1]  **Mukul Gagrani** [1]  **Mingu Lee** [1]  **Chris Lott** [1]

## Abstract

Large Language Models (LLMs) possess latent multi-token prediction (MTP) abilities despite being trained only for next-token generation. We introduce ESP (Embedding-Space Probing), a simple and **training-free** MTP method that probes an LLM using **on-the-fly mask tokens** drawn from its embedding space, enabling parallel future-token prediction without modifying weights or relying on draft models. We construct a speculative token tree by sampling Top-$K$ candidates from mask-token logits and apply a lightweight pruning rule to retain high-probability continuations. During generation, predictions are verified in parallel, yielding lossless decoding while significantly reducing model calls and increasing token throughput. ESP consistently outperforms existing training-free baselines, improving acceptance length by $7-11\%$ over Lookahead Decoding on `LLaMA3` and $7-8\%$ on `Qwen3`, and increasing throughput by up to $15-19\%$ over the strongest baseline. Finally, we provide theoretical insight and empirical evidence showing that decoder layers naturally align mask-token representations with next-token states, enabling accurate multi-step predictions **without retraining or auxiliary models**.

## 1. Introduction

Recent work in LLM inference has explored **multi-token prediction (MTP)** as a way to better utilize GPU parallelism, reduce latency, and accelerate generation. Traditional autoregressive decoding generates one token per step, leaving substantial compute underutilized. MTP methods instead aim to predict multiple future tokens in parallel (Gloeckle et al., 2024). However, many existing approaches rely on training auxiliary heads, modifying base model weights, or employing external draft models, as commonly seen in speculative decoding frameworks (Cai et al., 2024; Chen et al., 2023; Leviathan et al., 2023).

Despite the effectiveness of these approaches, training even small auxiliary models requires significant engineering effort, including dataset construction, architecture tuning, and days of GPU compute (Cottier et al., 2024; Goel et al., 2024). Moreover, these methods introduce additional parameters and memory overhead—for example, Cai et al. (2024) add LM heads of size ~400M parameters for `LLaMA3.2-3B-Instruct` (Dubey et al., 2024), while (Li et al., 2024) adds additional draft decoder layers—which makes them unsuitable for **edge devices** and compute-constrained environments. In contrast, training-free methods offer plug-and-play operation with frozen models, **require no retraining**, and **generalize across architectures and tasks while preserving lossless generation**.

We depart from prior MTP approaches that modify training objectives such as multi-head future prediction (Cai et al., 2024)), introduce additional heads or inference-time markers, such as PaSS (Monea et al., 2023), or are primarily diagnostic rather than algorithmic (e.g., Future Lens (Pal et al., 2023)). Instead, we present ESP, a **training-free, single-model, probing-based** approach that synthesizes mask tokens directly in the embedding space to elicit multi-token distributions from a frozen model. These parallel proposals are organized into a dynamic draft tree and pruned using a simple rule that optimizes acceptance under fixed block complexity, enabling efficient and lossless decoding without auxiliary models.

In this paper, we introduce a training-free, plug-and-play MTP method that works with any frozen LLM. ESP builds on a simple but powerful idea: **probing the model's internal generative capacity** using **on-the-fly generated mask tokens**. These tokens, synthesized in the model's embedding space and injected into the prompt, elicit predictions of multiple future tokens in parallel. The resulting predictions are then jointly verified by the base model, enabling efficient and lossless decoding.

To structure these predictions, we use a dynamic token-tree expansion mechanism that adaptively grows token paths based on cumulative probabilities. A lightweight pruning step removes duplicated or low-probability paths, improv-

[1] Qualcomm AI Research. Correspondence to: Raghavv Goel, Mukul Gagrani, Mingu Lee <{raghgoel,mgagrani,mingul}@qti.qualcomm.com>.

ing efficiency while maintaining diversity among parallel proposals. ESP supports multiple mask-token initialization strategies, including mean-of-prompt embeddings and sampling from the token embedding space; empirically, we find that mean-prompt initialization performs best across different model families.

To ensure practical scalability, we develop an efficient implementation of tree-attention masks and positional index updates that dramatically reduces runtime overhead, yielding up to $26\%$ higher token throughput than standard decoding for `LLaMA3.1-8B-Instruct`.

We evaluate our method on SpecBench (Xia et al., 2024), a diverse benchmark covering summarization, translation, reasoning, coding, and mathematical tasks, using both `LLaMA3` and `Qwen3` (Yang et al., 2025a). ESP consistently outperforms training-free baselines such as Lookahead Decoding (Fu et al., 2024) and Prompt Lookup Decoding (Saxena, 2023), achieving higher acceptance rates, fewer forward passes, and improved token throughput across tasks, different sampling temperatures, and model families.

Finally, we conduct quantitative and qualitative studies of token acceptance behavior, illustrating how acceptance varies with mask-token design and task type. Our method demonstrates strong performance across both open-ended tasks (writing, roleplay) and constrained tasks (summarization, math, reasoning), and is particularly suitable for compute-limited settings such as edge devices.

We make the following contributions:

1. **Training-free multi-token prediction via probing:** We introduce ESP (Embedding-Space Probing), a novel MTP paradigm that uses mask-token probing in the base model's embedding space, enabling multi-token generation without retraining or external draft models.

2. **Dynamic tree expansion for flexible decoding:** We propose a dynamic speculative token-tree expansion mechanism that adaptively grows based on predicted token probabilities, removing the need for manually designed tree structures.

3. **Efficient static-tree implementation:** We present a GPU-friendly implementation of static tree attention masks and position updates that significantly improves throughput for fixed structures.

4. **Theoretical and empirical justification:** We prove that alignment between mask-token and true-token representations (via cosine similarity) guarantees inclusion of the correct token in Top-$K$ predictions, and provide empirical evidence showing how this alignment emerges across layers.

5. **Comprehensive evaluation on SpecBench:** We perform extensive experiments demonstrating consistent improvements across models, tasks, and draft-tree sizes.

## 2. Background

We consider a frozen autoregressive language model $f_\theta$ with parameters $\theta$. Given a prompt sequence $x_{1:t}$, the model produces next-token logits and distribution:

$$f_\theta(x_{1:t}) \in \mathbb{R}^V, \qquad x_{t+1} \sim \text{softmax}(f_\theta(x_{1:t})). \quad (1)$$

To enable *multi-token prediction* without modifying model parameters, we inject **mask tokens** $m_1, m_2, \ldots, m_k$ into the prompt. These tokens are computed dynamically from the model's embedding space and appended as:

$$x_{1:t}, \ m_1, m_2, \ldots, m_k.$$

Each mask token $m_i$ is designed to elicit a prediction of a future token $x_{t+1+i}$. We use pairs of mask tokens $(m_{s+i}^1, m_{s+i}^2)$ that share parameters across positions but attend to different contexts via a causal tree attention mask. After inserting mask tokens, the model outputs future-token predictions at the mask positions:

$$\hat{x}_{t+1+i} \sim \text{softmax}(f_\theta(x_{1:t}, m_1, m_2, \ldots, m_k)[t+1+i]). \quad (2)$$

**Verification (Speculative-Decoding Style).** To ensure correctness, predicted tokens are verified against the base model's own next-token distribution (simultaneous verification as in (Lin et al., 2025)):

$$\hat{x}_{t+1+i} \overset{?}{=} x_{t+1+i} \sim \text{softmax}(f_\theta(x_{1:t+i})). \quad (3)$$

If $\hat{x}_{t+2}$ is accepted, it is appended to the prefix and used to verify $\hat{x}_{t+3}$, and so on. This yields a *lossless* procedure consistent with speculative decoding and multi-token prediction (Fu et al., 2024; Leviathan et al., 2023), with the key difference that *ESP generates speculative futures via mask-token probing rather than a separate draft model*. We visualize vanilla autoregressive decoding, mask-token probing, and simultaneous verification in Figure 1.

We defer the discussion of *tree branching*—where multiple candidate tokens are considered per mask token—to Section 3. There, we introduce a dynamic token tree construction mechanism that expands token paths based on cumulative probabilities and includes pruning to improve diversity.

## 3. Methods

We propose ESP, a training-free multi-token prediction framework that probes frozen LLMs using dynamically

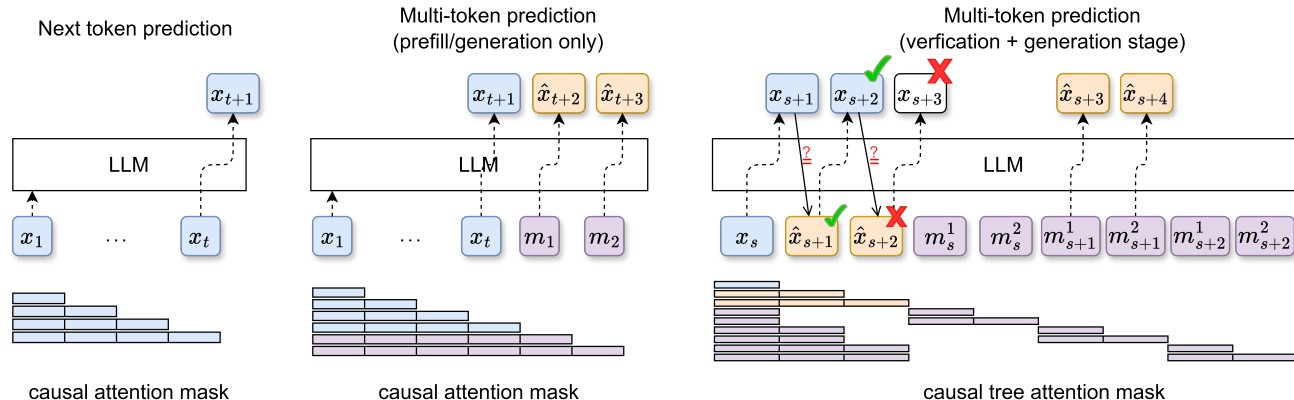

*Figure 1.* (Left) Standard next-token prediction setup for autoregressive models, (middle) multi-token prediction during prefill-stage by probing mask tokens which are appended to prompt tokens, (right) multi-token prediction with parallel verification and generation. Mask tokens are associated with last generated token ($x_s$) and future tokens ($\hat{x}_{s+1}, \hat{x}_{s+2}$) through custom tree attention mask.

generated mask tokens. These mask tokens are injected into the prompt and used to elicit predictions for multiple future tokens in a single forward pass. ESP doesn't require any auxiliary draft models or fine-tuning. The predicted tokens are verified sequentially using the base model itself, ensuring consistency with its autoregressive behavior. To support richer token exploration, we introduce a dynamic token tree construction mechanism that expands future token paths based on cumulative probabilities, along with a pruning strategy to eliminate redundant tokens. We also define block complexity as a key setting to trade-off parallelism and compute cost.

### 3.1. Mask Token Injection

Let the input prompt be a sequence of tokens $x_{1:t} \triangleq [x_1, x_2, \ldots, x_t]$. These tokens are first projected into the embedding space via the model's embedding matrix $E \in \mathbb{R}^{V \times d}$, where $V$ is the vocabulary size and $d$ is the embedding dimension: $\mathbf{e}_i = E[x_i], \ \forall, i = 1, \ldots, t$.

We initialize each of the $k$ mask tokens with the mean of the prompt embeddings:

$$m_i = \frac{1}{t} \sum_{j=1}^{t} \mathbf{e}_j, \quad \forall, i = 1, \ldots, k. \tag{4}$$

This soft, prompt-context-dependent initialization produces a mask embedding statistically aligned with the prompt distribution, which we find probes the frozen LLM more effectively.

During the generation phase, mask tokens are updated based on the tokens generated, adding more context-based information:

$$m_i[s+1] = m_i[s] + \lambda(\mathbf{e}_{t+s} - m_i[s]), \quad \forall, i \tag{5}$$

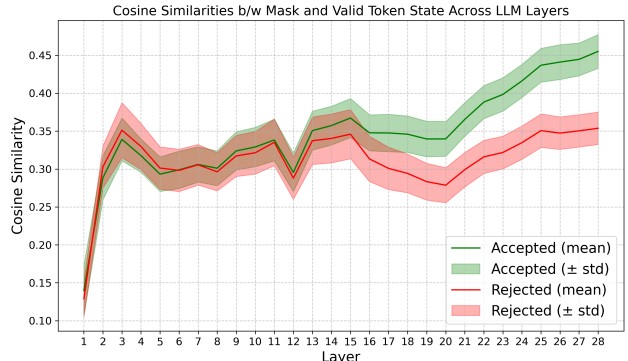

*Figure 2.* We use Dolly-Databricks (creative-writing) (Conover et al., 2023) samples (100) to measure average cosine similarity across layers for mask and true-future token hidden-states. For `Llama3.2-3B-Instruct`, higher cosine similarity in later layers (15 onwards) correlates with token acceptance (green), while lower similarity correlates with rejection (red).

where $s$ denotes the generation step and $\lambda$ is a positive scalar.

Although the mask tokens take the same embedding value across all token trajectories in the future token tree, their position IDs and past context differ, leading to diverse generations. We compare alternative initialization strategies — prompt-based hard initialization and embedding-distribution sampling — in Section 4.

### 3.2. Why Mask Tokens Enable Multi-Token Prediction

Our method relies on the observation that decoder layers progressively align the mask token representation, aligning its hidden state with that of valid tokens. This alignment is critical because the LM head computes logits by taking inner products between the final hidden state and its vocabulary columns $W_r \in \mathbb{R}^d$. Higher inner product results in a higher logit, increasing the likelihood that the token appears in the top-K candidates.

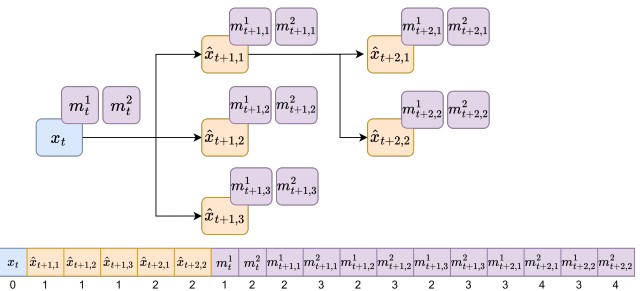

*Figure 3.* Mask tokens are present for each input token: last generated (blue) and future (orange) tokens, when processed by model all tokens are flattened and mask tokens are placed at the end with appropriate position indices.

To quantify this alignment, we track the evolution of cosine similarity between mask and next true token hidden states across layers. Specifically, we compare the hidden states of the next true token $x_{t+1}$ and the corresponding mask token $m_{t+1}$, both of which are used to predict the token at position t+2 (next-next true token). As shown in Figure 2, accepted tokens exhibit a steady increase in cosine similarity after layer 15, reaching an average of about $0.45$, while rejected tokens plateau near $0.35$. This divergence suggests that higher similarity correlates with token acceptance. We formalize this observation with the following lemma:

**Lemma 3.1.** *Let $h_m, h_v \in \mathbb{R}^d$ be hidden states for the mask token and the next-true token after the last decoder layer and let $W \in \mathbb{R}^{d \times V}$ be the LM head with columns $w_r \in \mathbb{R}^d$. Assume $||h_m||_2, ||h_v||_2 \leq c_h$ and $||w_r||_2 \leq c_w, \forall r$. We define $i^* = \arg\max_r w_r^T h_v$ as the next-next true token (under greedy sampling) and $S_m = \arg\text{topK}_r w_r^T h_m$ be the next top-K tokens under the mask token. Then,*

$$i^* \in S_m, \; if \; \cos(h_m, h_v) \geq \delta^* \; for \; some \; \delta^* \in \mathbb{R}_{>0} \quad (6)$$

i.e., the next-next true token (under greedy sampling) belongs to the set of top-$K$ draft tokens generated from mask token when cosine similarity between next-true and mask token states exceed $\delta^*$ threshold. The proof is provided in Appendix in Section A.

### 3.3. Logit-based prediction and verification

This section describes how ESP (i) proposes future tokens from mask-token logits, (ii) verifies those proposals using the base model, and (iii) controls compute through block complexity.

At each mask-token position, ESP uses the output logits to propose future tokens conditioned on the current context. Specifically, we sample Top-K candidates from each mask-token distribution, and these candidates form the nodes of a speculative token tree. The total number of sampled candidates across all depths determines the size of the draft tree explored by ESP.

During prefill, ESP performs only generation. In the decode phase, it performs both generation and verification, as shown in Figure 1. We expand only the Top-1 token at each depth (details in Appendix Section D), which keeps computation bounded while favoring high-likelihood continuations.

Verification is lossless: each predicted token is checked against the base model's own next-token prediction and is accepted only if it matches exactly. Once a token is accepted, the corresponding mask token(s), aligned to that accepted position through the tree-attention mask, are used to generate the next set of future-token proposals (Fig. 3). In this way, ESP alternates between proposing multiple future continuations and verifying them with the same frozen model.

Because verification evaluates a bundle of speculative tokens in a single forward pass, the computation can become expensive when the bundle grows large. We therefore define block complexity as the total number of tokens processed in parallel in one forward pass. This includes: (i) the last accepted token, (ii) all draft-tree nodes corresponding to predicted future tokens, and (iii) the associated mask tokens used to probe those future positions.

Example (two mask tokens, depth-two tree). Suppose we use two mask tokens per future step and sample K1 and K2 tokens at depths 1 and 2, respectively. Then the block contains: 1 last-accepted token, $K_1 + K_2$ predicted future tokens, and, $2(1 + K_1 + K_2)$ mask tokens. Therefore, the block complexity is:

$$\text{Block Complexity} = 3\big(1 + K_1 + K_2\big),$$

In the general case, with k mask tokens and sample sizes Ki at depth i, the block complexity is:

$$\text{Block Complexity} = (k + 1)\left(1 + \sum_{i=1}^{k} K_i\right). \quad (7)$$

We compare all methods under matched block complexity, since larger trees can improve acceptance but also increase latency and compute cost.

### 3.4. Dynamic Tree Construction

Constructing a tree of future token predictions typically requires selecting a fixed Top-$K$ from each mask token's logits. However, this approach is brittle and task-dependent, often requiring extensive tuning across models and domains. Instead, we propose a dynamic draft tree construction method that adapts to the model's uncertainty by using cumulative probability to prioritize high-likelihood future token trajectories.

Our tree follows a **Top-1** expansion strategy, where only the highest-probability token at each depth is expanded to form

---

**Algorithm 1** Dynamic Generation of Token Tree

---

1: **Input:** LLM $M$, Budget $B$, Mask count $k$, Logits $l_{m_{1:k}}$, Context $x$
2: **Output:** Draft token tree $\mathcal{T}$ with $B$ nodes
3: Initialize $\mathcal{T}$ with root $r$, $P(r) \leftarrow 1.0$ ▷ Root probability
4: $\mathcal{N} \leftarrow \{r\}, \mathcal{C} \leftarrow \emptyset$        ▷ $\mathcal{N}$: nodes, $\mathcal{C}$: candidates
5: **for** $i = 1$ to $k$ **do**
6:      $\mathcal{C}_{\text{new}} \leftarrow \emptyset$
7:      **for all** $n \in \mathcal{N}$ at depth $i - 1$ **do**
8:          Get logits $l_n$ from $M$ given path of $n$
9:          Sample top-$(B-i)$ tokens $\{t_1, \ldots, t_{B-i}\}$ from $l_n$
10:          **for all** token $t_j$ in sampled tokens **do**
11:             Create child $c$ with token $t_j$ appended to path of $n$
12:             $P(c) \leftarrow P(n) \cdot P(t_j|l_n)$    ▷ Update probability
13:             $\mathcal{C}_{\text{new}} \leftarrow \mathcal{C}_{\text{new}} \cup \{c\}$
14:          **end for**
15:      **end for**
16:      $\mathcal{C} \leftarrow \mathcal{C} \cup \mathcal{C}_{\text{new}}$
17:      **if** $i < k$ **then**
18:          $\mathcal{N} \leftarrow \text{top}_{B-i}(\mathcal{C})$      ▷ Sort by probability
19:          $\mathcal{C} \leftarrow \mathcal{C} \setminus \mathcal{N}$
20:      **end if**
21: **end for**
22: Add top $(B - 1)$ nodes from $\mathcal{C}$ to tree $\mathcal{T}$
23: **Return** $\mathcal{T}$

---

child nodes. This design simplifies the tree while preserving efficiency, as illustrated in Appendix Figure 6. We leave exploration of more complex tree structures to future work. After computing cumulative probabilities for all token trajectories, we select the Top-$B-1$ trajectories, where $B$ is the block complexity (including the last generated token). This allows the tree to grow adaptively: more branches are explored when the model is uncertain, and fewer when it is confident.

Our algorithm, shown in Algorithm 1, takes as input the block complexity (budget) and the number of mask tokens (tree depth), and outputs a set of token trajectories that maximize coverage while respecting the computational budget. This avoids exhaustive grid search over tree branch (Top-$K$) and and ensures that the tree structure adapts to the model's predicted probabilities rather than relying on fixed heuristics. We show in Section 4 that dynamic tree generation performs on-par or better than hand-crafted tree branches.

### 3.5. Tree Pruning

To reduce redundancy during tree expansion, we apply a simple pruning heuristic which removes consecutive repeated tokens—for example, when a child node predicts the same

token as its parent (e.g., parent = "the", child = "the"). We observe that mask token predictions often include the last generated token or the parent token, which is typically redundant. To address this, we replace such token(s) with the next best token candidate from the mask token output distribution.

We perform ablation on using tree pruner and provide observations in Appendix Section G.7 that tree pruner helps improve avg token acceptance by up to $4\%$.

## 4. Experiments

To evaluate the efficacy of ESP, we conduct rigorous experiments using latest open-source frontier models: (a) `LLaMA3`, and (b) `Qwen3`, We use sample matching. where a token is accepted only if it is an exact match, enabling **lossless generation**.

**Models**: We evaluate two LLaMA3 models— `LLaMA3.2-3B-Instruct` and `LLaMA3.1-8B-Instruct`—and two Qwen3 models—`Qwen3-8B` and `Qwen3-32B`—to demonstrate that our method generalizes across architectures and scales. All models are run with a maximum generation length of 100 tokens on a single NVIDIA A100 GPU. We use temperature=$0.0, 1.0$ with temperature=$1.0$ results provided in the Appendix Section G.3.

**Tasks**: We use tasks from **SpecBench** Xia et al. (2024), which includes summarization, translation, writing, coding, retrieval and math tasks (from GSM8K Cobbe et al. (2021)).

**Baselines**: We compare against training-free and draft-free speculative decoding methods: (i) **Prompt Lookup Decoding (PLD)** (Saxena, 2023), (ii) **Stochastic Adaptive N-gram Drafting (STAND)** (Song et al., 2025), and (iii) **Lookahead Decoding (LADE)** (Fu et al., 2024). The configuration for baseline methods is mentioned in Appendix Section F based on their respective papers.

**Performance Metric**: We report **average acceptance length ($\tau$)**, defined as the average number of tokens accepted and the bonus token per model call. $\tau$ directly reflects the reduction in model calls: model calls $\propto \frac{1}{\tau}$, thus, **higher $\tau$ implies fewer model calls** and lower compute (energy) cost. We also report **Speedup Ratio (S/R)** to show the absolute wall-time on H100 GPUs.

**Block Complexity (BC)**: Number of draft tree nodes; three block complexities: $10, 30, 60$.

**Mask token design** in ESP is based on mean of given prompt's embedding (soft initialization) with dynamic updates based on the last token generated following Equation (5), with $\lambda = 0.1$. We use single mask token for BC=$10,30$ and two mask tokens for BC=$60$, unless other-

wise stated.

## 4.1. Results

We begin by reporting the average accepted tokens ($\tau$) and speedup ratio (S/R) over autoregressive decoding on Spec-Bench for two block complexities: BC = 30 and BC = 60. Across all models and budgets, ESP delivers the strongest overall speed–quality trade-off, achieving the best or tied-best throughput while maintaining lossless generation.

As shown in Table 1, our method (ESP) consistently out-performs existing baselines, achieving up to **12% higher** $\tau$ **than LADE** on LLaMA3 models and **13–18% higher** $\tau$ **than STAND** on Qwen3 models. This translates to a substantial reduction in the number of forward model calls, up to **42% fewer model invocations** at BC = 60, as detailed in Appendix Section G.1 in Table 6. ESP achieves the best speedup ratios across all models and budgets, out-performing the next-best baseline by up to $\sim$ 12% on LLaMA3.1-8B-Instruct (over PLD) and 13–22% on Qwen3 models (over LADE), while crucially **not relying on any auxiliary N-gram cache**. These gains stem from ESP's ability to generate higher-quality speculative continuations directly from the base model, which leads to longer accepted token chains during verification and therefore fewer model invocations overall.

We further present comprehensive average tokens accepted and speed-up ratio (S/R) results across all downstream tasks and block complexities (BC = 10, 30, 60) for LLaMA3.1-8B-Instruct and Qwen3-32B, shown in Figure 4 and Figure 5. Each method is color-coded, with higher opacity indicating larger block complexity. Our method, ESP (green) consistently achieves the highest $\tau$ and speedup across most tasks and BC settings, demonstrating that **LLMs, when probed effectively, can predict future tokens across diverse tasks**. For LLaMA3.1-8B-Instruct, STAND (blue) performs second-best on coding, RAG, and summarization — tasks with high n-gram repetition where tokens can be directly copied from the prompt — while LADE (orange) is second-best on translation and math/reasoning. A similar task-dependent pattern is observed for Qwen3-32B, where LADE increasingly competes with STAND as BC grows.

For LLaMA3.1-8B-Instruct, the coding task yields the highest overall $\tau$ (e.g., STAND BC=30: $\tau$=2.27), driven by the n-gram copy advantage, while our method achieves its peak on math/reasoning ($\tau$=1.81, 1.43$\times$ speedup at BC=60). For Qwen3-32B, math/reasoning yields the highest $\tau$ gains across all methods, with our method reaching $\tau$=1.75 (1.53$\times$ speedup at BC=60). Importantly, our method performs well even at **low BC** (avg 1.22$\times$ at BC=10 for LLaMA3.1-8B-Instruct vs 1.23$\times$ for STAND), making it suitable for **edge devices** where com-

pute constraints limit block size. Exceptions include coding, RAG, and summarization on LLaMA3.1-8B-Instruct, where STAND outperforms our method due to its n-gram copy advantage. Additional block efficiency results on LLaMA3.2-3B-Instruct and Qwen3-8B at BC=60 are shown in Appendix Section G.2, Figure 8. Qualitative results of ESP are shown in Appendix Section G.10.

> **Key takeaway**: at small block complexities (e.g., BC = 10, 30), ESP with a single mask token already surpasses existing baselines by a clear margin. This supports the hypothesis that frozen LLMs, when probed appropriately, can reliably predict additional future tokens with minimal overhead

## 4.2. Dynamic Tree Expansion and number of mask tokens to probe

To evaluate the effectiveness of our dynamic tree expansion, we perform an ablation study comparing different branching strategies when using two mask tokens ($m_1, m_2$). Results for BC = 30 and BC = 60 are shown in Table 2. We observe that dynamic branching consistently performs best for both BC=30 and BC=60 avoiding the need for optimal branch configuration search.

The size and structure of tree width (branches) are constrained by the number of mask tokens and the block complexity. For a single mask token, we use the maximum possible tree branch since only one future token is predicted. For example, with BC = 10 and 30, the tree branches are [4] and [14], respectively—no dynamic branching is needed in this case. For two mask tokens, multiple branching configurations are possible. For BC = 30, the tree can branch as $[9 - i, i]$ for $i \in 1, \ldots, 8$, allowing up to 9 branches. Similarly, for BC = 60, the tree can branch as $[19 - i, i]$ for $i \in 1, \ldots, 18$. Our dynamic tree expansion avoids exhaustive search over these configurations by adaptively selecting the optimal branching based on the input prompt, yielding strong BE performance.

Importantly, the number of tree branches tends to be smaller when more mask tokens are used, due to the need to respect the block complexity constraint. This trade-off can impact average accepted tokens, and our dynamic expansion strategy helps navigate it effectively.

We also ablate over number of mask tokens, since these models are trained for predicting only the next token, increasing the number of mask tokens can lead to diminishing returns. In Table 3 we compare different numbers of mask tokens with fixed BC=60 for Llama3 models. We observe that one and two mask configuration perform better than three mask configuration.

In Appendix Section G.6, we provide a comprehensive table

*Table 1.* Comparison of multi-token prediction **average acceptance length** ($\tau$) and **speedup ratio** (S/R) over autoregressive decoding across models and methods averaged on Spec-bench tasks for block complexity $BC = 30$ and $BC = 60$. All results on a single NVIDIA H100 GPU with 256-token outputs. PLD uses a fixed depth-10 tree (BC≈10) regardless of budget; its values are shown for reference under both BC columns.

| Model | BC=30 | | | | | | | | BC=60 | | | | | | | |
|---|---|---|---|---|---|---|---|---|---|---|---|---|---|---|---|---|
| | PLD | | STAND | | LADE | | ESP | | PLD | | STAND | | LADE | | ESP | |
| | $\tau$ | S/R | $\tau$ | S/R | $\tau$ | S/R | $\tau$ | S/R | $\tau$ | S/R | $\tau$ | S/R | $\tau$ | S/R | $\tau$ | S/R |
| LLaMA3.2-3B-Instruct | 1.43 | 1.19× | 1.55 | 1.01× | 1.43 | 1.01× | **1.56** | **1.22×** | 1.43 | 1.19× | 1.62 | 1.07× | 1.57 | 1.10× | **1.63** | **1.22×** |
| LLaMA3.1-8B-Instruct | 1.44 | 1.23× | 1.58 | 1.10× | 1.45 | 1.06× | **1.63** | **1.35×** | 1.44 | 1.23× | 1.64 | 1.14× | 1.60 | 1.14× | **1.71** | **1.38×** |
| Qwen3-8B | 1.31 | 1.12× | 1.42 | 1.02× | 1.55 | 1.12× | **1.63** | **1.37×** | 1.31 | 1.12× | 1.48 | 1.06× | 1.73 | 1.21× | **1.74** | **1.43×** |
| Qwen3-32B | 1.29 | 1.09× | 1.42 | 1.10× | 1.52 | 1.22× | **1.60** | **1.43×** | 1.29 | 1.09× | 1.48 | 1.13× | 1.69 | 1.31× | **1.70** | **1.48×** |

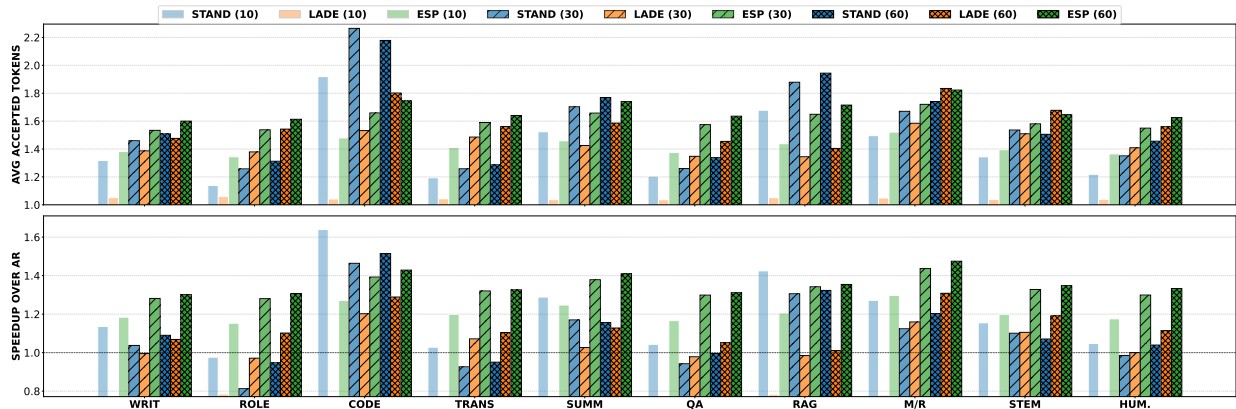

*Figure 4.* Evaluation on Spec-Bench using `LLaMA3.1-8B-Instruct` across block complexities (BC = 10, 30, 60). Our method (green) consistently achieves the highest average accepted tokens across most tasks and BC settings. WRIT=writing, ROLE=roleplay, CODE=coding, TRANS=translation, SUMM=summarization, M/R=math/reasoning, HUM.=humanities

comparing $\tau$ across tasks for both configurations. We observe that open-ended tasks (e.g., writing, reasoning) tend to perform better with a single mask token—benefiting from longer future token sequences and wider tree (greater exploration)—while closed-ended tasks (e.g., translation, math) perform better with two mask tokens deeper and less-wide tree (more focused and efficient exploitation of the model's predictions).

### 4.3. Efficient tree attention mask and position index construction

Tree-based decoding requires attention masks and position indices that respect branching hierarchies, which traditionally involves sequential iteration over tree nodes — a process that is **not GPU-friendly** and incurs high latency. To address this, we implement an **efficient strategy** that caches the attention mask and incrementally appends columns as new tokens are accepted, avoiding recomputation. Similarly, position indices are updated via a simple offset, enabling fast reuse across generation steps. These optimizations are detailed in Appendix Section E.

This approach significantly improves throughput for fixed tree structures. As shown in Table 4, our efficient implementation consistently improves throughput across all configurations, with average gains of ∼22% for `LLaMA3.2-3B-Instruct` and ∼21%

for `LLaMA3.1-8B-Instruct` over the naive implementation. Gains are especially pronounced at higher block complexities, with improvements of 29–30% for `LLaMA3.1-8B-Instruct` and 27–28% for `LLaMA3.2-3B-Instruct` at BC = 60 (Table 4).

### 4.4. Different mask embedding designs
We compare three strategies for initializing the mask token embedding:

a. Last K (hard init) — uses the embeddings of the last $k$ prompt tokens, $m_i = \mathbf{e}_{t-k-i}, \ \forall, i = 1, \ldots, k$

b. Sample — draws each mask token from a Gaussian fit to the full embedding table, $m_i \sim \mathcal{N}(\mu, \sigma^2 I)$, where $\mu$ and $\sigma$ are the mean and standard deviation across all $V$ vocabulary embeddings

c. Mean (soft init) — the prompt-mean strategy used in our main method Equation (4)

Among the variants tested, the mean of the prompt embeddings consistently yields the best performance across LLaMA3 models, as shown in Table 5. This initialization provides a strong contextual prior, allowing the model to better align its predictions with the prompt semantics. These results suggest that even simple embedding-based heuristics

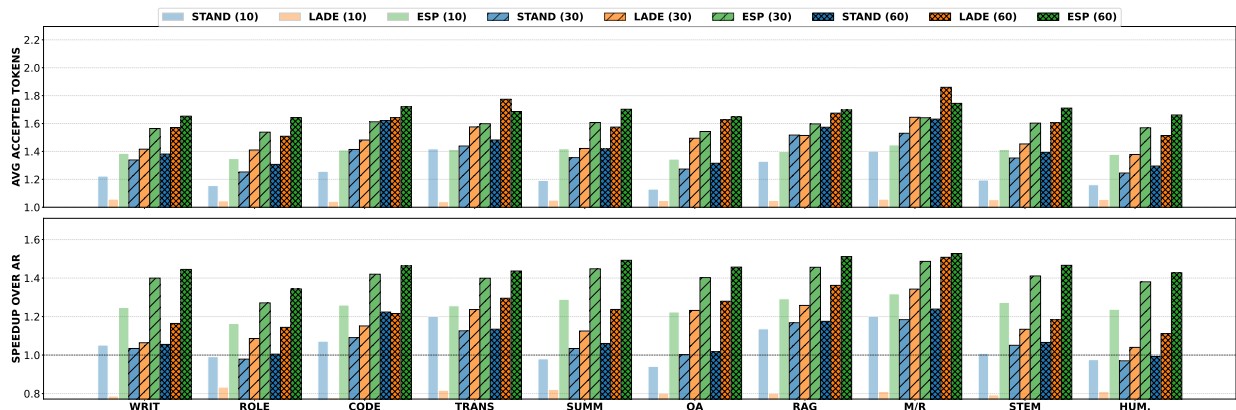

*Figure 5.* Evaluation on SpecBench using `Qwen3-32B` across block complexities (BC = 10, 30, 60). Our method (green) consistently achieves the highest average accepted token across most tasks and BC settings.

*Table 2.* Average acceptance length ($\tau$) for BC= 30 and BC= 60 for `LLaMA3.2-3B`/`LLaMA3.1-8B-Instruct` with two mask tokens ($m_1, m_2$) across different branch configurations. Dynamic uses adaptive branching per input; static configs use fixed trees. The best $\tau$ is **bold**; second best is underlined.

| Model | **BC=30** ($m_1, m_2$) | | | | **BC=60** ($m_1, m_2$) | | | | | |
| --- | --- | --- | --- | --- | --- | --- | --- | --- | --- | --- |
| | dynamic | [7,2] | [5,4] | [3,6] | dynamic | [15,4] | [12,7] | [10,9] | [8,11] | [6,13] |
| `LLaMA3.2-3B-Instruct` | **1.506** | 1.504 | 1.476 | 1.417 | **1.630** | 1.631 | 1.620 | 1.602 | 1.574 | 1.540 |
| `LLaMA3.1-8B-Instruct` | **1.571** | 1.570 | 1.540 | 1.479 | **1.712** | 1.708 | 1.701 | 1.687 | 1.665 | 1.624 |

*Table 3.* Impact of number of mask tokens at BC=60. Branch widths are set in decreasing order across mask tokens. **Bold** = best $\tau$.

| Model | ($m_1$) [29] | ($m_1, m_2$) [15, 4] | ($m_1, m_2, m_3$) [7, 5, 3] |
| --- | --- | --- | --- |
| `LLaMA3.2-3B-Instruct` | **1.65** | 1.63 | 1.51 |
| `LLaMA3.1-8B-Instruct` | **1.73** | 1.71 | 1.57 |

*Table 4.* Speedup over AR for naive and efficient implementations across mask configurations single and two mask tokens. For two mask tokens ($m_1, m_2$): [7, 2] for BC=30 and [15, 4] for BC=60.

| Method | $m_1$(30) | $m_1, m_2$(30) | $m_1$(60) | $m_1, m_2$(60) |
| --- | --- | --- | --- | --- |
| | `LLaMA3.2-3B-Instruct` | | | |
| $\tau$ | 1.56 | 1.50 | 1.65 | 1.63 |
| Naive | 0.99× | 0.99× | 0.95× | 0.96× |
| Efficient | **1.22×** | 1.08× | 1.20× | **1.22×** |
| | `LLaMA3.1-8B-Instruct` | | | |
| $\tau$ | 1.63 | 1.57 | 1.73 | 1.71 |
| Naive | 1.14× | 1.08× | 1.05× | 1.07× |
| Efficient | **1.35×** | 1.16× | 1.36× | **1.38×** |

can significantly influence multi-token prediction quality in training-free settings. We additionally run experiments to stress-test the robustness of the mask token embedding design in Appendix Section G.8 and observed a minor performance drop when the mask token initialization is outside the embedding table distribution.

We additionally present ablations on the impact of increasing block complexity (BC = 30 to 120) in Appendix Section G.4, and a comparison of speedup ratio with training-based method in Appendix Section G.5.

## 5. Related Work

**Mechanistic Interpretability** Future Lens (Pal et al., 2023) demonstrates that future token information is linearly decodable from single hidden states and via single-state interventions, providing a mechanistic rationale for our probing strategy. In parallel (Wu et al.), formalize whether transformers plan ahead, offering pre-caching versus breadcrumbs hypotheses consistent with our empirical finding that hidden states can be mined to propose several future tokens.

**Multi-token prediction (MTP)** accelerates LLM inference by predicting multiple tokens in parallel (Gloeckle et al., 2024; Guo et al., 2025). Recent methods include MEDUSA (Cai et al., 2024), which adds decoding heads and tree-based attention, and masked-input approaches with learnable sampler modules (Samragh et al., 2025). Other works train independent output heads atop a shared trunk (Gloeckle et al., 2024; Mehra et al., 2025) while finetuning the base model parameters. Unlike these, our method is training-free and operates on frozen LLMs using mask token probing and dynamic tree construction.

**Speculative Decoding (SD)** generates draft tokens for par-

*Table 5.* $\tau$ for different mask token initializations for LLaMA3.2-3B-Instruct and LLaMA3.1-8B-Instruct. Last K sets $m_i = \mathbf{e}_{t-k-i}$, Sample draws $m_i \sim \mathcal{N}(\mu, \sigma^2 I)$ where $\mu, \sigma$ are the mean and standard deviation of the full embedding table, and Mean (used in our main method).

| Method | LLaMA3.2-3B-Instruct | | | LLaMA3.1-8B-Instruct | | |
|---|---|---|---|---|---|---|
| | $m_1(10)$ | $m_1(30)$ | $m_1, m_2(60)$ | $m_1(10)$ | $m_1(30)$ | $m_1, m_2(60)$ |
| Last K (hard init) | 1.36 | 1.53 | 1.62 | 1.38 | 1.56 | 1.67 |
| Sample (embedding distribution) | 1.39 | 1.57 | 1.65 | 1.41 | 1.60 | 1.69 |
| Mean (soft init) | **1.41** | **1.59** | **1.67** | **1.42** | **1.62** | **1.71** |

allel verification (Leviathan et al., 2023; Chen et al., 2023). Extensions improve block efficiency via token trees (Miao et al., 2023; Sun et al., 2023). Some reuse target model layers for drafting (Zhang et al., 2024; Elhoushi et al., 2024), while others modify attention for multi-token prediction (Bhendawade et al., 2024; Lin et al., 2025). These approaches often require architectural changes or auxiliary models; ours does not.

**Training-free acceleration** includes Swift (Xia et al.), Jacobi decoding (Santilli et al., 2023), LADE (Fu et al., 2024), and PLD/STAND (Saxena, 2023; Song et al., 2025). These methods rely on caching or heuristic matching, whereas our approach avoids memory overhead and uses probing for dynamic tree construction.

Additional related work, using prompt/register tokens for MTP (Chen et al., 2024; Gerontopoulos et al., 2026), are discussed in Appendix Section B.

## 6. Conclusion

We present a training-free framework for multi-token prediction via probing, leveraging mask tokens from the embedding space to guide parallel generation. Our method introduces dynamic token tree expansion and block complexity-aware decoding, enabling efficient speculative inference without requiring draft models, N-gram caches, or offline token trees. Our theoretical insight reasons why probing with mask tokens leads to correct token prediction. Through extensive experiments across diverse tasks and model scales, we demonstrate that even a single mask token can yield substantial gains in block efficiency leading to reduction in model forward calls—often outperforming existing baselines—while dynamic branching with multiple mask tokens further enhances performance in deeper tree configurations. Our analysis reveals that open-ended tasks benefit from wider token trees, while closed-ended tasks prefer deeper, more focused trees. We also propose efficient implementations for attention masks and position IDs that significantly reduce runtime overhead. These findings collectively support the hypothesis that large language models, when probed appropriately, can confidently predict multiple future tokens without additional training or architectural changes.

## Impact Statement

This work introduces a training-free multi-token prediction framework that enables substantial inference-time speedups for large language models without requiring model retraining, auxiliary networks, or architectural modifications. By leveraging mask-token probing and dynamic token-tree expansion, our method delivers consistent gains in throughput and efficiency across model families while maintaining lossless generation. These properties make the approach particularly impactful for compute-constrained environments such as mobile devices, embedded systems, and edge deployments, where storage, memory, and power budgets limit the use of heavyweight speculative decoding frameworks. Beyond practical efficiency improvements, the method sheds light on latent multi-step predictive structure within decoder representations, offering insights for interpretability and future model design. We anticipate that this work will help broaden access to high-quality LLM inference by lowering deployment costs and enabling faster, more scalable generation in real-world applications.

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

## A. Intuition for Multi-Token Prediction Using Mask Tokens

This section provides the formal proof for the claim stated in Section 3.2 of the main paper, which involves comparing the hidden states of the next-true token and the forecast (mask) token when both attempt to predict the same position. As an example, for a prompt with $n$ tokens, we compare the hidden states of:

$$x_{t+1} \text{ (next-true token)}, \quad m_{t+1} \text{ (mask token)}$$

both trying to predict the $(t+2)^{\text{th}}$ token:

$$x_{t+2}, \quad \hat{x}_{t+2,i} \text{ for } i \in \text{argtopK}_i \text{ logits}_{m_{t+1}}.$$

This analysis is only possible in hindsight, as during inference we do not have access to next-true token at the current generation step, and hope to generate next-next true using mask token.

We show theoretically that a higher cosine similarity between the hidden states of the mask token and the next-true token results in the next-next-true token being included in the Top-$K$ prediction set of the mask token. Intuitively, we believe that later decoder layers in LLMs enrich the mask token representation so that it aligns with the valid token state, thereby improving the likelihood of predicting correct next-next token.

**Lemma A.1.** *Let $h_m, h_v \in \mathbb{R}^d$ be hidden states for the mask token and the next-true token after the last decoder layer and let $W \in \mathbb{R}^{d \times V}$ be the LM head with columns $w_r \in \mathbb{R}^d$. Assume $||h_m||_2, ||h_v||_2 \leq c_h$ and $||w_r||_2 \leq c_w, \forall r$. We define $i^* = \text{argmax}_r w_r^T h_v$ as the next-next true token (under greedy sampling) and $S_m = \text{argtopK}_r w_r^T h_m$ be the next top-K tokens under the mask token. Then,*

$$i^* \in S_m, \text{ if } cosine\_similarity(h_m, h_v) \geq \delta^* \text{ for some } \delta^* \in \mathbb{R}_{>0}$$

*Proof.* For each $r$ we have:
$$|w_r^\top (h_m - h_v)| \leq ||w_r||_2 ||h_m - h_v||_2 \leq c_w ||h_m - h_v||_2 \tag{8}$$

Now compute:
$$||h_m - h_v||_2^2 = ||h_m||^2 + ||h_v||^2 - 2h_m^\top h_v.$$

Using $cosine\_similarity(h_m, h_v) \geq \delta$:
$$h_m^\top h_v \geq ||h_m|| ||h_v|| \delta \geq c_h^2 \delta.$$

So:
$$||h_m - h_v||_2^2 \leq 2c_h^2 - 2c_h^2 \delta = 2c_h^2 (1 - \delta).$$

Therefore:
$$||h_m - h_v||_2 \leq \sqrt{2} c_h \sqrt{1 - \delta}$$

Combining with equation 8 we get
$$|w_r^\top (h_m - h_v)| \leq c\sqrt{1 - \delta}, \tag{9}$$

where $c = \sqrt{2} c_w c_h$. Now, define $\Delta_j^m = w_{i^*}^T h_m - w_j^T h_m$ and $\Delta_j^v = w_{i^*}^T h_v - w_j^T h_v$ be the difference in the logits between the token $i^*$ and $j$ under the mask token and next-true token respectively. We know that $\Delta_j^v \geq 0, \forall j$ since $i^* = \arg\max_r w_r^T h_v$. Also, using equation 9 we have:

$$w_{i^*}^T h_m - w_{i^*}^T h_v \geq -c\sqrt{1 - \delta}$$
$$w_j^T h_v - w_j^T h_m \geq -c\sqrt{1 - \delta}$$

Adding the above two inequalities we get:

$$\Delta_j^m - \Delta_j^v \geq -2c\sqrt{1 - \delta}$$
$$\implies \Delta_j^m \geq \Delta_j^v - 2c\sqrt{1 - \delta}$$

Hence, $\Delta_j^m > 0$ if $\Delta_j^v - 2c\sqrt{1-\delta} > 0$ which holds true when

$$\delta > 1 - \left(\frac{\Delta_j^v}{2c}\right)^2 \tag{10}$$

i.e., token $j$ has smaller logit value than token $i^*$ even under mask token output logits if Equation (10) holds.

Let $S_v \triangleq \operatorname{argtopK}_r w_r^T h_v$ be the top-K tokens under the valid token $v$, and $K^*$ be the top-$K^{th}$ index for $w_r^T h_v$ under next-true token $v$ as well. Suppose $\delta > \delta^*$ where $\delta^* = 1 - \left(\frac{\Delta_{K^*}^v}{2c}\right)^2$, then we have the following:

$$\delta > 1 - \left(\frac{\Delta_{K^*}^v}{2c}\right)^2 \geq 1 - \left(\frac{\Delta_j^v}{2c}\right)^2, \forall j \notin S_v \tag{11}$$

This is because $\Delta_{K^*}^v \leq \Delta_j^v, \forall j \notin S_v$ due to the fact that the $K^*$ is in the top-K token set $S_v$.

Therefore using Equation (10), 11 we get,

$$\Delta_j^m = w_{i^*}^T h_m - w_j^T h_m \geq 0, \forall j \notin S_v$$
$$\implies w_{i^*}^T h_m \geq w_j^T h_m, \forall j \notin S_v$$

Hence, there are at least $V - K$ tokens for which $\Delta_j^m > 0$. Therefore, $i^*$ is in the top-K set $S_m$ when $\delta > \delta^* = 1 - \left(\frac{\Delta_{K^*}^v}{2c}\right)^2$

$\square$

Note that to match the top-1 next-next true token with mask token outputs, $\Delta_{i^*}^v = 0 \implies \delta^* = 1$, i.e., $\cos(h_m, h_v) = 1$. Thus, as the size of top-K set increases a smaller value of $\delta^*$ can still result in valid matches. To the best of our knowledge, this is the first proof to show connections between cosine similarity and acceptance in multi-token setting.

Finally, understanding why mask tokens achieve higher cosine similarity—specifically, how relevant information is injected into mask token states in the first place—remains an interesting question, which we leave for future work.

## B. Extended Related Works

**Multi-token prediction history** – The idea of predicting multiple tokens has a long history: ProphetNet (Qi et al., 2020) pre-trains with **future n-gram** prediction, while debates around NTP's limitations have advocated teacherless multi-token objectives; and PaSS (Monea et al., 2023) proposes **parallel speculative sampling** with a single model via a marker token. Our work is **training-free**, **head-free**, and focuses on acceptance-aware parallel proposals via **embedding mask probing**.

**Multi-token prediction** – Multi-token prediction (MTP) has emerged as a promising direction for accelerating LLM inference (Gloeckle et al., 2024) by leveraging parallelism and improving sample efficiency (Guo et al., 2025). MEDUSA (Cai et al., 2024) augments LLMs with multiple decoding heads and a tree-based attention mechanism to predict future tokens in parallel, while fine-tuning entire model. Samragh et al. (2025) introduce a masked-input formulation with gated LoRA adaptation and a learnable sampler module, enabling multi-token generation. (Gloeckle et al., 2024) trains a model to predict multiple future tokens using independent output heads atop a shared trunk. Couple more methods (Chen et al., 2024), (Gerontopoulos et al., 2026) add learnable prompt/register tokens to base model to enable multi-token predictions, where similar idea has been proposed apriori in Lin et al. (2025). Chen et al. (2024) involves training prompt tokens for each different model, and uses a static tree, whereas (Gerontopoulos et al., 2026) involves updating weights of the base model. Unlike these methods, our approach is entirely training-free and operates on any frozen LLMs, using mask token probing and dynamic tree construction to achieve efficient multi-token prediction without architectural changes or extra parameter overheads.

**Speculative Decoding** (SD) – SD accelerates LLM inference by generating draft tokens using a smaller or faster model, which are then verified by the target model in parallel (Leviathan et al., 2023; Chen et al., 2023). Several extensions improve block efficiency by generating token trees (Miao et al., 2023; Sun et al., 2023; Jeon et al., 2024; Yang et al., 2025b; Li

et al., 2024). More recently, methods eliminate the need for a separate draft model by reusing target model layers. For example, (Zhang et al., 2024) skips layers to derive a draft model with adaptive exit, while (Elhoushi et al., 2024) uses early-exit verification trained with layer dropout. (Bhendawade et al., 2024) replaces multi-head attention in the final layers with multi-stream attention to predict multiple tokens concurrently, requiring end-to-end training. (Lin et al., 2025) introduces learnable mask and prompt tokens, enabling tree-based decoding and verification in a single forward pass. While effective, these methods often require architectural changes or auxiliary drafter models, unlike our approach, which is entirely training-free and drafter-free.

**Training-free Acceleration** – A growing body of work explores training-free approaches to accelerate autoregressive decoding. Swift (Xia et al.) proposes a training-free variant of (Zhang et al., 2024) by selecting draft layers online via context-aware Bayesian optimization. Jacobi decoding (Santilli et al., 2023) formulates parallel decoding as a system of non-linear equations solved through fixed-point iteration, but consistently performs worst to LADE. LADE (Fu et al., 2024) tracks token trajectories using a fixed-size 2D window and generates n-grams in parallel, which are later verified by the target model. Prompt Lookup Decoding (PLD) (Saxena, 2023) matches the last generated n-gram to past context and, if matched, uses the subsequent token trajectory for verification. STAND (Song et al., 2025) extends PLD by storing logits of the entire past context and sampling a token tree from matched n-gram strings. Compared to LADE and STAND, our method avoids memory overhead from caching logits or token trajectories.

# C. Future Directions

**Mechanistic Interpretability**: Recent work in mechanistic interpretability (Conmy et al., 2023) has focused on probing internal components of language models, such as linear layers and activation patching. While our probing-based multi-token prediction (MTP) method is not directly aligned with these approaches, we believe it offers a complementary perspective. Specifically, our use of mask tokens from the embedding space opens up new avenues for understanding LLM behavior. Future work could explore how different types of mask tokens interact with model internals, potentially revealing interpretable structures or activation patterns that guide multi-token generation.

**Combining Baseline Decoding Methods with MTP**: Our method can be synergistically combined with existing decoding strategies such as Lookahead Decoding (LADE) and Prompt-Lookup Decoding to further improve block efficiency and reduce the number of forward passes. One promising direction is to fuse token trajectories from both the N-gram cache in LADE and our probing-based method, then select top candidates from the combined pool. This hybrid approach could yield more robust and efficient decoding, especially in long-context or high-throughput settings.

**Efficient Dynamic Tree Attention Implementation**: In our current implementation, using multiple mask tokens with dynamic token trees results in varying attention masks across model passes. A naive solution is to precompute and store attention masks for all possible branch configurations, but this quickly becomes memory-intensive as block complexity (BC) increases. Future work could explore more efficient attention computation strategies, such as dynamic masking via sparse attention kernels or runtime mask synthesis, to reduce memory overhead while preserving flexibility. Additionally, integration with efficient inference frameworks such vLLM will also be explored, where tree-attention based speculative decoding method such as EAGLE-2 (Li et al., 2024) are already integrated.

**Principled Design of Mask Tokens**: Our current work explores simple instantiations of mask tokens, but a more principled approach to mask token design could yield significant gains in block efficiency. For instance, control-theoretic or optimization-based methods to select mask configurations, may lead to more effective probing. Investigating the geometry and distribution of mask tokens in embedding space could also provide insights into their predictive power and generalization behavior.

**Combining with Diffusion LLMs (dLLMs)**: Recently, diffusion-based language models (dLLMs) have gained popularity for their competitive performance compared to autoregressive LLMs (AR-LLMs). These models are inherently trained using mask tokens with a bidirectional attention mask, a fundamental difference from the causal attention mask used in AR-LLMs. Several efficient designs have been proposed (Sahoo et al., 2024; Wu et al., 2025; Israel et al., 2026). In future, it would be worth exploring designs that combine our method with bidirectional masked models capable of predicting multiple tokens by default.

## D. Token Tree Construct Details

We perform **Top-1 tree expansion**, meaning that at each depth of the token tree, only the token with the highest probability is expanded to generate child nodes, following the approach in (Lin et al., 2025). This strategy becomes relevant when using two or more mask tokens.

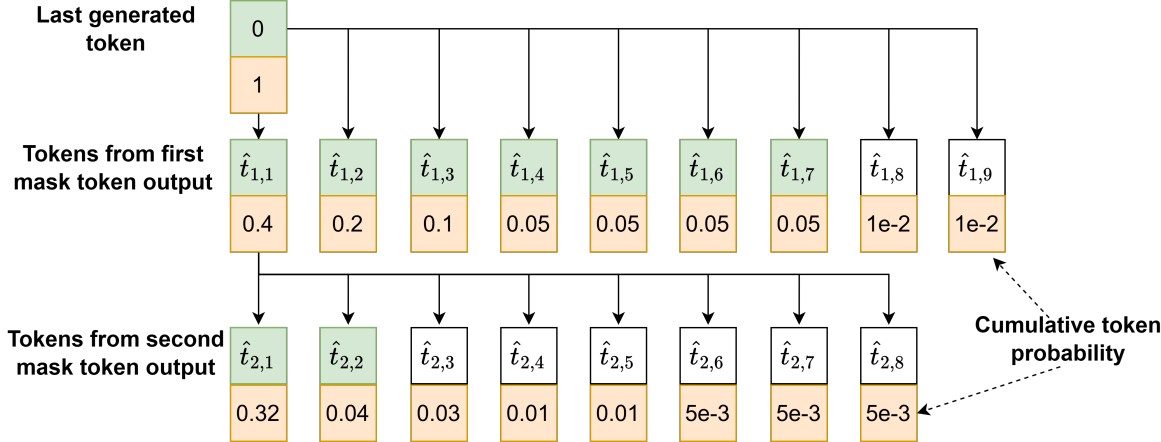

*Figure 6.* Example of dynamic token tree expansion for block complexity (BC) = 30 using two mask tokens. As the tree expands, each child node inherits the probability of its parent, resulting in a multiplicative score. We denote the last accepted token (root) as 0, the Top-9 tokens from the output logits of the first mask token as $\hat{t}_{1,1:9}$, and the Top-8 tokens from the second mask token as $\hat{t}_{2,1:8}$.

Our motivation for this choice stems from the observation that generations from mask tokens $(m_2, \ldots, m_k)$ exhibit **weak conditional dependence**. Let the output probability distribution of the LLM for input $x_{t-1}$ be defined as:

$$p_\theta(x_{t-1} \mid x_{<t-1}) \tag{12}$$

The output distributions from mask tokens are then:

$$p_\theta(m_1 \mid x_{t-1}, x_{<t-1}) \rightarrow 1^{\text{st}} \text{ mask token output} \tag{13}$$

$$p_\theta(m_2 \mid m_1, x_{t-1}, x_{<t-1}) \rightarrow 2^{\text{nd}} \text{ mask token output} \tag{14}$$

$$p_\theta(m_k \mid m_{k-1}, \ldots, m_1, x_{t-1}, x_{<t-1}) \rightarrow k^{\text{th}} \text{ mask token output}$$

From the second mask token onward, the conditional generation includes previous mask tokens—serving as an approximation of actual token embeddings. As these mask tokens propagate through the model, their hidden states are refined, eventually predicting the correct token.

Expanding only the Top-1 token at each depth effectively **maximizes the approximate joint likelihood** of generation from multiple mask tokens. This expansion also helps maintain the token tree within a manageable block complexity (BC). Dynamic tree expansion is used to prioritize high-likelihood trajectories while avoiding exponential growth in tree size.

An example is shown in Figure 6 for two mask tokens $(m_1, m_2)$ with BC = 30. In this case, we retain the Top-7 tokens from the output logits of $m_1$ and the Top-2 tokens from $m_2$, resulting in a compact yet expressive tree structure.

## E. Efficient Tree Attention Mask and Position ID Construction

Using a token tree structure requires respecting its branching hierarchy during input construction, which introduces a sequential loop over tree nodes. This affects how position IDs and attention masks are computed—each must reflect the structure of the tree and the order of tokens in the input. In our proposed method, we found that custom tree attention mask generation is a sequential process that iterates over all branches of the token tree, resulting in high latency.

To address this, we exploit the static structure of the token tree—specifically when using a single mask token or two mask tokens with fixed branching—during the simultaneous generation and verification phase. As described in **??**, for a given

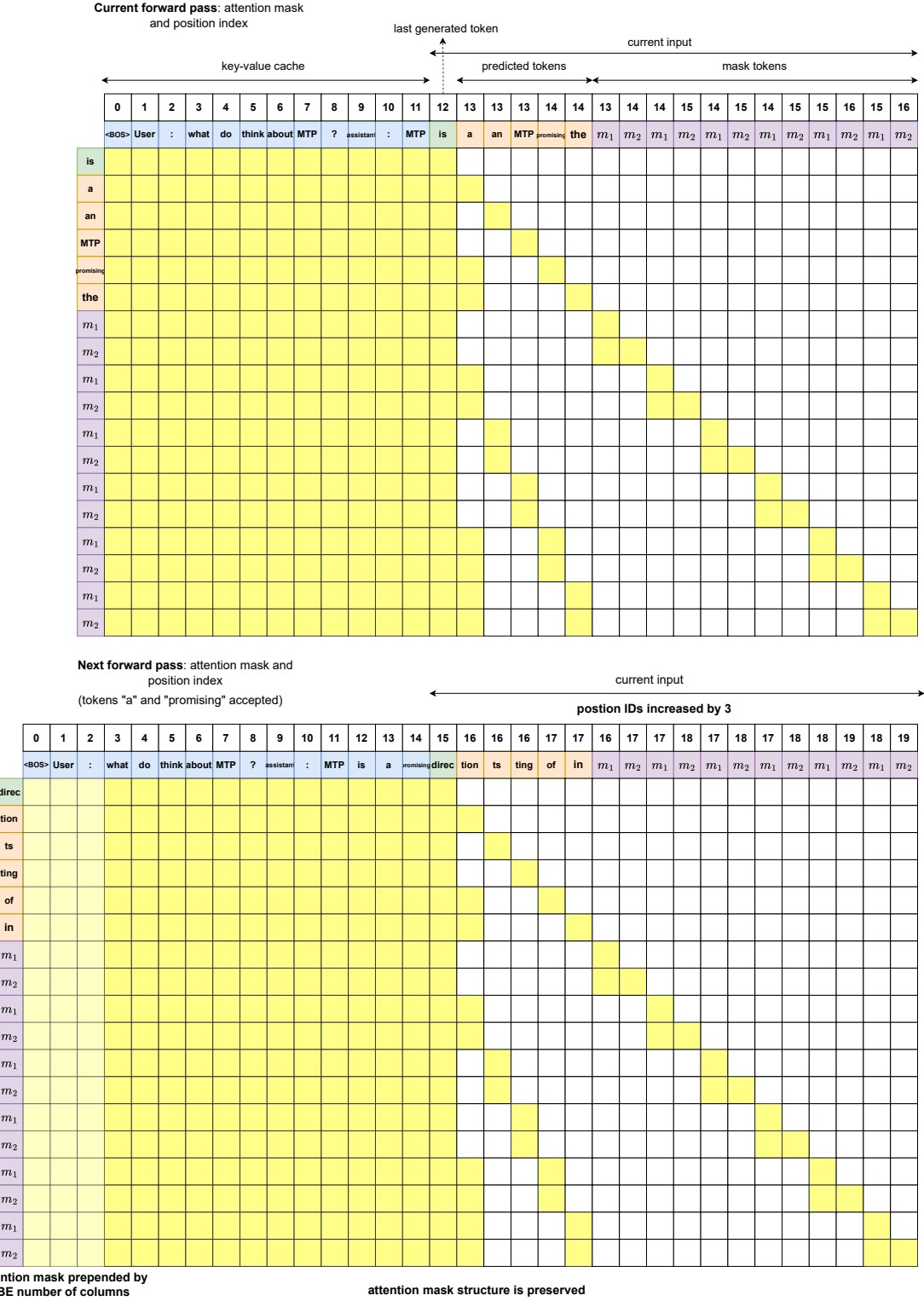

*Figure 7.* Attention mask and position index for the current (top) and next (bottom) LLM forward pass in our method with static branch configuration [3,2]. The tokens in blue are part of key-value cache, green token is the last generated token which will be inputted to the model to generate next token, orange tokens are future tokens, and purple tokens are mask tokens. Position IDs of each token is mentioned above it. The structure of the attention mask remains unchanged except for the addition of BE columns filled with zeros (assuming non-attended positions are represented by $-\infty$). Similarly, the position indices are preserved in order but uniformly incremented by BE.

block complexity (BC), the input token order is fixed: last generated token, future predicted tokens, and mask tokens. Each of these components requires custom position indices (PID) and tree attention masks, as illustrated in Figure 1.

After the prefill phase, as generation progresses, both the PID tensor and attention mask evolve based on the number of tokens generated (i.e., block efficiency, BE) as shown in Figure 7. We observe that:

**Position IDs (PID)**: The only change across generation steps is a uniform shift in indices. The PID tensor from the previous step can be reused by simply adding the number of generated tokens (BE) to each index.

**Attention Mask**: The shape of the attention mask changes as the key-value cache grows. Specifically, the number of columns increases by BE, and these new columns are filled with zeros (assuming non-attended positions are represented by $-\infty$). This effectively prepends BE columns of zeros to the previous step's mask, yielding the correct attention mask for the current pass.

In Table 4 in main manuscript, we report token-rate improvements for token trees with varying number of mask tokens. For example, using a single mask token (with tree node counts of 15 and 30 for BC = 30 and 60, respectively), we observe: for LLaMA3.2-3B-Instruct, token-rate increases by $4\%$ for BC = 30 and $19.6\%$ for BC = 60. For LLaMA3.1-8B-Instruct, token-rate increases by $28\%$ for BC = 30 and $22.4\%$ for BC = 60. These results demonstrate that efficient reuse of attention masks and position IDs can significantly accelerate decoding, especially for larger token trees and larger models.

Note that, token-rate is influenced by both block efficiency (BE) and future token tree size, which depend on block complexity and the number of mask tokens. Our method scales well with tree depth and enables efficient decoding for high-capacity models. Future work will explore GPU-optimized dynamic tree expansion and integration with serving frameworks like vLLM (Kwon et al., 2023) and SGLang (Zheng et al., 2024).

# F. Baseline Configurations

**PLD**: Uses a token tree of depth 10, with one token per depth. The future token trajectory is selected by matching the last generated n-gram to past context and verifying the subsequent tokens.

**STAND**: Constructs a token tree of depth 10, with additional branches at each depth based on block complexity. The number of tokens per depth is determined by selecting candidates with the highest cumulative probability from stored logits of matched n-gram strings. STAND is equivalent to PLD for smaller BC=10.

**LADE**: Defines block complexity as $(L-1)(W+G)$

- $L$: N-gram size
- $W$: Window size
- $G$: Guess set size

Configurations used:

- BC=10: $L$=3, $W$=4, $G$=1 (minimum window size required is 3)
- BC=30: $L$=4, $W$=5, $G$=5
- BC=60: $L$=5, $W$=8, $G$=7

# G. Additional Results

We provide additional comprehensive results to offer deeper insights into the behavior and performance of the proposed method.

### G.1. Reduction in model forward calls

Our method achieves the highest reduction in model forward passes, as shown in Table 6. Reducing the number of model calls helps save compute and energy, which is especially beneficial for edge or portable devices.

*Table 6.* Comparison of multi-token prediction **reduction in model forward calls** performance across models and methods averaged on Spec-bench tasks for block complexity $BC = 30$ and $BC = 60$

| Model | BC=30 | | | | BC=60 | | | |
|---|---|---|---|---|---|---|---|---|
| | **PLD** | **STAND** | **LADE** | **ESP** | **PLD** | **STAND** | **LADE** | **ESP** |
| LLaMA3.2-3B-Instruct | 30.07 | 35.48 | 30.07 | **35.90** | 30.07 | 38.27 | 36.31 | **38.65** |
| LLaMA3.1-8B-Instruct | 30.56 | 36.71 | 31.03 | **38.65** | 30.56 | 39.02 | 37.50 | **41.52** |
| Qwen3-8B | 23.66 | 29.58 | 35.48 | **38.65** | 23.66 | 32.43 | 42.20 | **42.53** |
| Qwen3-32B | 22.48 | 29.58 | 34.21 | **37.50** | 22.48 | 32.43 | 40.83 | **41.18** |

## G.2. Block Efficiency on additional models

We also compare BE across downstream tasks for LLaMA3.2-3B-Instruct and Qwen3-8B at BC = 60, as shown in Figure 8a and Figure 8b. Similar to results in Figure 4, 5, our method outperforms other baselines across most tasks, with the exception of 'retrieval' on LLaMA3.2-3B-Instruct and 'summarization' on QWen3-8B, where it performs second best. For Qwen3 models, single mask token is used for BC=60 while for LLaMA3 model two mask tokens are used.

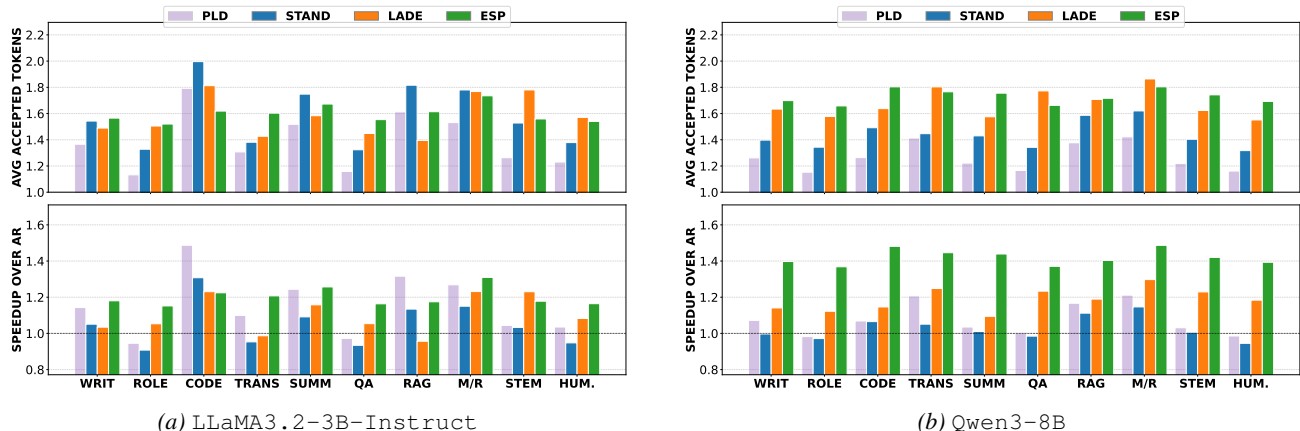

*(a)* LLaMA3.2-3B-Instruct          *(b)* Qwen3-8B

*Figure 8.* Block efficiency across SpecBench tasks for LLaMA3.2-3B-Instruct and QWen3-8B at block complexity (BC) = 60. Our method consistently outperforms baselines across most tasks, demonstrating strong performance in both open-ended and closed-ended settings.

## G.3. Impact of Sampling

We run all methods with base model in sampling mode with **temperature=1.0** for BC=30, 60 with output length = 100, and provide the block efficiency in for Llama3.2-3B-Instruct and Llama3.1-8B-Instruct in Table 7, 8 respectively. Our method outperforms other methods in terms of maximum number of mean accepted tokens on average across different SpecBench tasks showing the efficacy of our method for different sampling strategies.

*Table 7.* Block-efficiency with **temperature=**1.0 for Llama3.2-3B-Instruct across SpecBench tasks for BC = 30 and BC = 60. Task acronyms: WRIT (writing), ROLE (roleplay), CODE (coding), TRANS (translation), SUMM (summarization), QA (question answering), RAG (retrieval-augmented generation), M/R (math/reasoning).

| Method | BC | WRIT | ROLE | CODE | TRANS | SUMM | QA | RAG | M/R | AVG |
|---|---|---|---|---|---|---|---|---|---|---|
| PLD | 30 | 1.53 | 1.10 | 1.24 | 1.22 | 1.62 | 1.15 | 1.77 | 1.46 | 1.39 |
| STAND | 30 | 1.31 | 1.17 | 1.31 | 1.20 | 1.55 | 1.18 | 1.53 | 1.43 | 1.33 |
| LADE | 30 | 1.37 | 1.27 | 1.39 | 1.33 | 1.38 | 1.39 | 1.39 | 1.52 | 1.38 |
| OURS | 30 | 1.51 | 1.46 | 1.53 | 1.52 | 1.57 | 1.44 | 1.49 | 1.61 | **1.52** |
| STAND | 60 | 1.33 | 1.22 | 1.33 | 1.22 | 1.61 | 1.22 | 1.58 | 1.46 | 1.37 |
| LADE | 60 | 1.49 | 1.38 | 1.56 | 1.42 | 1.50 | 1.47 | 1.55 | 1.64 | 1.50 |
| OURS | 60 | 1.60 | 1.53 | 1.63 | 1.60 | 1.65 | 1.54 | 1.59 | 1.70 | **1.61** |

*Table 8.* Block-efficiency with **temperature=**1.0 for `LLama3.1-8B-Instruct` across tasks for BC = 30 and BC = 60. Task acronyms: WRIT (writing), ROLE (roleplay), CODE (coding), TRANS (translation), SUMM (summarization), QA (question answering), RAG (retrieval-augmented generation), M/R (math/reasoning).

| Method | BC | WRIT | ROLE | CODE | TRANS | SUMM | QA | RAG | M/R | AVG |
|--------|-----|------|------|------|-------|------|------|------|------|------|
| PLD | 30 | 1.40 | 1.12 | 1.25 | 1.20 | 1.51 | 1.14 | 1.48 | 1.40 | 1.31 |
| STAND | 30 | 1.24 | 1.14 | 1.34 | 1.25 | 1.56 | 1.17 | 1.50 | 1.39 | 1.32 |
| LADE | 30 | 1.22 | 1.16 | 1.31 | 1.28 | 1.26 | 1.21 | 1.25 | 1.32 | 1.25 |
| OURS | 30 | 1.52 | 1.48 | 1.59 | 1.56 | 1.61 | 1.53 | 1.55 | 1.65 | **1.56** |
| STAND | 60 | 1.26 | 1.17 | 1.35 | 1.28 | 1.60 | 1.21 | 1.56 | 1.47 | 1.36 |
| LADE | 60 | 1.25 | 1.22 | 1.38 | 1.37 | 1.41 | 1.28 | 1.34 | 1.46 | 1.34 |
| OURS | 60 | 1.61 | 1.57 | 1.70 | 1.65 | 1.70 | 1.63 | 1.64 | 1.74 | **1.66** |

## G.4. Impact of Increasing Block Complexity

*Table 9.* Impact of number of mask tokens on average acceptance length ($\tau$). Each additional mask doubles the budget: BC=30 ($m_1$), BC=60 ($m_1, m_2$), BC=120 ($m_1, m_2, m_3$). All configs use static tree expansion.

| Model | $m_1$ (BC=30) | $m_1, m_2$ (BC=60) | $m_1, m_2, m_3$ (BC=120) |
|-------|------|--------|---------|
| `LLaMA3.2-3B-Instruct` | 1.561 | 1.631 | 1.669 |
| `LLaMA3.1-8B-Instruct` | 1.632 | 1.708 | 1.756 |

We observe that increasing the number of mask tokens with BC leads to diminishing gains, as seen in Table 9. This is because autoregressive models are trained only to predict the next token, and therefore predicting further future tokens becomes increasingly harder.

## G.5. Comparison with Training-based Method

*Table 10.* Speedup ratio (S/R) over autoregressive decoding. EAGLE3 requires 200–300 GPU hours of training; all other methods are training-free (0 GPU hours). N/A = no official EAGLE3 checkpoint available.

| Model | BC | PLD | STAND | LADE | Ours | GPU hrs (training) | EAGLE3 | GPU hrs (training) |
|-------|-----|------|-------|------|------|------|--------|------|
| `LLaMA-3.1-8B-Instruct` | 30 | 1.24 | 1.10 | 1.06 | **1.35** | 0 | 2.36 | ~200–300 |
| | 60 | 1.24 | 1.15 | 1.14 | **1.38** | | 2.61 | |
| `Qwen3-32B` | 30 | 1.07 | 1.07 | 1.17 | **1.43** | 0 | N/A (no official checkpoint) | |
| | 60 | 1.07 | 1.11 | 1.24 | **1.48** | | | |

We compare training-free multi-token prediction algorithms with the state-of-the-art training-based method, EAGLE3 (Li et al., 2026), in Table 10. The trade-off is clear: EAGLE3 requires 200–300 GPU hours of training per model (training time can increase for larger model sizes), whereas training-free methods can be used in a plug-and-play manner with no additional compute cost.

## G.6. Impact of Number of Mask Tokens

We observe that different downstream tasks benefit from different numbers of mask tokens, as shown in Table 12 for BC = 60. Tasks such as 'roleplay' and 'question answering' — more open-ended in nature — achieve higher block efficiency with a single mask token (shallow and wider tree). In contrast, tasks like 'coding', 'summarization', 'RAG', and 'math/reasoning' — more closed-ended — perform better with two mask tokens (deeper and more focused tree). 'Writing' shows mixed behavior: $m_1$ is preferred for `LLaMA3.2-3B-Instruct` while $m_1, m_2$ is marginally better for `LLaMA3.1-8B-Instruct`. 'Translation' is largely tied across both configurations (difference $\leq 0.01$).

More specifically, for BC = 60, probing with a single mask token $m_1$ results in a token tree size of 30 (shallow and wider), whereas using two mask tokens ($m_1, m_2$) yields a tree size of 20 (deeper and less wide).

For smaller BC = 30, using a single mask token consistently outperforms two mask tokens, as shown in Table 11. The two-mask configuration uses our dynamic tree expansion algorithm. We consider performance to be similar when the BE difference is within 0.01.

*Table 11.* Block-efficiency across different tasks for **BC=30** with different mask tokens probed (single mask: $m_1$, two mask tokens: $m_1, m_2$). Task acronyms are WRIT: writing, ROLE: roleplay, CODE: coding, TRANS: translation, SUMM: summarization, M/R: math/reasoning

| Config | WRIT | ROLE | CODE | TRANS | SUMM | QA | RAG | M/R |
|---|---|---|---|---|---|---|---|---|
| Llama3.2-3B-Instruct | | | | | | | | |
| $m_1$ | 1.56 | 1.54 | 1.57 | 1.56 | 1.64 | 1.55 | 1.70 | 1.69 |
| $m_1, m_2$ | 1.52 | 1.51 | 1.53 | 1.53 | 1.59 | 1.50 | 1.69 | 1.67 |
| Llama3.1-8B-Instruct | | | | | | | | |
| $m_1$ | 1.58 | 1.56 | 1.62 | 1.62 | 1.65 | 1.59 | 1.73 | 1.70 |
| $m_1, m_2$ | 1.54 | 1.45 | 1.58 | 1.59 | 1.62 | 1.52 | 1.71 | 1.68 |

*Table 12.* Block-efficiency across different tasks for **BC=60** comparing different mask tokens probed. Task acronyms are WRIT: writing, ROLE: roleplay, CODE: coding, TRANS: translation, SUMM: summarization, M/R: math/reasoning

| Config | WRIT | ROLE | CODE | TRANS | SUMM | QA | RAG | M/R |
|---|---|---|---|---|---|---|---|---|
| Llama3.2-3B-Instruct | | | | | | | | |
| $m_1$ | **1.64** | **1.65** | 1.64 | 1.65 | 1.72 | **1.64** | 1.78 | 1.78 |
| $m_1, m_2$ | 1.62 | 1.61 | **1.67** | 1.64 | **1.74** | 1.62 | **1.83** | **1.81** |
| Llama3.1-8B-Instruct | | | | | | | | |
| $m_1$ | 1.67 | **1.64** | 1.73 | 1.70 | 1.74 | **1.69** | 1.80 | 1.78 |
| $m_1, m_2$ | **1.69** | 1.60 | **1.75** | 1.71 | 1.75 | 1.66 | **1.87** | **1.83** |

### G.7. Impact of tree-pruner

We evaluate the impact of our lightweight tree-pruner—which introduces no additional computational overhead—on block efficiency across SpecBench tasks for `LLaMA3.2-3B-Instruct` and `LLaMA3.1-8B-Instruct`, as shown in Table 13. The pruner is particularly effective when using two mask tokens, improving BE by up to 3% for `LLaMA3.2-3B-Instruct` and up to 4% for `LLaMA3.1-8B-Instruct`. For the single mask token setting, the pruner maintains BE without degradation.

This empirical result suggests that the **Top-$K$ predictions from the second mask token often include the Top-1 token from the first mask**, leading to redundancy. Our pruner removes such repeated tokens during tree expansion, enabling more diverse branching without sacrificing accuracy.

*Table 13.* Impact of tree pruning for different LLaMA models

| Method | $\mathbf{m_1(10)}$ | $\mathbf{m_1(30)}$ | $\mathbf{m_1, m_2(30)}$ | $\mathbf{m_1, m_2(60)}$ |
|---|---|---|---|---|
| LLaMA3.2-3B-Instruct | | | | |
| w/o tree pruning | 1.41 | 1.59 | 1.50 | 1.63 |
| w/ tree pruning | 1.41 | 1.59 | **1.55** | **1.67** |
| LLaMA3.1-8B-Instruct | | | | |
| w/o tree pruning | 1.42 | 1.62 | 1.51 | 1.66 |
| w/ tree pruning | 1.42 | 1.62 | **1.57** | **1.71** |

### G.8. Impact of initializing mask tokens outside embedding distribution

To emphasize the robustness of mask token initialization, we compare soft initialization with sampling (as described in Section 3.1) against initializing the mask token using a high standard deviation ($\mu + 5\sigma, \mu + 10\sigma$), where $\mu$ and $\sigma$ denote the **mean and standard deviation of the entire embedding table**. We observe that block efficiency is only slightly affected, indicating that mask token initialization is robust across different settings. This further suggests that LLMs possess an inherent ability to predict future tokens.

*Table 14.* `Llama3.2-3B-Instruct` average block efficiency (BE) across Spec-bench categories with BC=60. Task acronyms are WRIT: writing, ROLE: roleplay, CODE: coding, TRANS: translation, SUMM: summarization, M/R: math/reasoning

| Method | WRIT | ROLE | CODE | TRANS | SUMM | QA | RAG | M/R | AVG |
|---|---|---|---|---|---|---|---|---|---|
| $\mu + \epsilon\sigma$ | 1.60 | 1.58 | 1.65 | 1.61 | 1.72 | 1.59 | 1.64 | 1.79 | **1.65** |
| $\mu + 5\sigma$ | 1.56 | 1.55 | 1.60 | 1.60 | 1.67 | 1.55 | 1.61 | 1.75 | **1.61** |
| $\mu + 10\sigma$ | 1.54 | 1.55 | 1.61 | 1.60 | 1.69 | 1.55 | 1.61 | 1.75 | **1.61** |

### G.9. Impact of Mask Token Update Rate

*Table 15.* Impact of EMA update rate $\lambda$ for mask token embeddings on average acceptance length ($\tau$) at BC=30 and BC=60. Default $\lambda = 0.1$ consistently performs best; higher $\lambda$ degrades performance more than lower $\lambda$, suggesting the method is robust to slower updates but sensitive to over-adaptation.

| Model | BC | $\lambda = 0.01$ | $\lambda = 0.1$ (default) | $\lambda = 0.5$ |
|---|---|---|---|---|
| `LLaMA3.2-3B-Instruct` | 30 | 1.53 | **1.59** | 1.43 |
| | 60 | 1.63 | **1.67** | 1.52 |
| `LLaMA3.1-8B-Instruct` | 30 | 1.60 | **1.62** | 1.47 |
| | 60 | 1.69 | **1.71** | 1.57 |

The EMA update rate $\lambda$ controls how quickly the mask token embedding adapts to the current context. As shown in Table 15, the default $\lambda = 0.1$ consistently achieves the best $\tau$ across both models and block complexities. A smaller $\lambda = 0.01$ (slower adaptation) incurs only a minor drop of $\sim$0.02–0.04, while a larger $\lambda = 0.5$ (faster adaptation) degrades performance more significantly ($\sim$0.10–0.15), suggesting the mask token benefits from stable, smoothed updates rather than rapidly tracking individual token statistics.

### G.10. Qualitative Examples of Token Accepted

To qualitatively evaluate the effectiveness of our probing-based multi-token prediction, we use the Dolly-Databricks dataset (Conover et al., 2023) focused on creative writing. This allows us to inspect which future tokens are correctly predicted by the model without any training, simply through probing with mask tokens.

Below is a generation from `LLaMA3.2-3B-Instruct` when probed with two mask tokens under block complexity (BC) = 60. The model is able to predict several future tokens (blue) with just simple mask token probing—without any training or any N-gram cache.

**Prompt:**
```
<|begin_of_text|><|start_header_id|>user<|end_header_id|>
How do you make the best shepherds pie?
<|eot_id|><|start_header_id|>assistant<|end_header_id|>
```

Shepherd's pie, a classic British dish that's easy to make and always a crowd-pleaser! Here's a recipe for the best shepherd's pie, with a few tips and tricks to ensure it's a true culinary masterpiece:

**Ingredients:**

For the filling:

* 1 pound ground lamb or beef (or a combination of both)
* 1 onion, diced
* 2 cloves of garlic, minced
* 1 cup frozen peas and carrots
* 1 cup beef broth
* 2 tablespoons tomato paste
* 2 tablespoons all-purpose flour
* 2 teaspoons Worcestershire sauce
* 1 teaspoon dried thyme
* Salt and pepper, to taste

For  the mashed potato topping:

* 3-4 large potatoes, peeled and chopped into 1-inch cubes
* 1/4 cup milk or half-and-half
* 2 tablespoons butter
* Salt and pepper, to taste

**Instructions:**
1.**Preheat your oven to 350°F (180°C).**
2.**Make the filling:**

* In a large pan over medium-high heat, cook the ground lamb or beef until browned,breaking it up witha spoon as it cooks.
* Add the diced onion and cook until softened, about 5 minutes.
* Add the minced garlic and cook for 1 minute.
* Stir inthe frozen peas and carrots, beef broth, tomato paste, flour, Worcestershire sauce, and thyme. Bring the mixture to a boil, then reducethe heat to low and simmer for 10-15 minutes, stirring occasionally.
* Season the filling with salt and pepper to taste.

3.**Make the mashed potato topping :**

* Boil the chopped potatoes in a large pot of salted water until they're tender, about 15-20 minutes.
* Drain the potatoes and return them to the pot. Add the milk, butter, salt, and pepper, an dmash the potatoes with a potato masher or a fork until they're smoothand creamy.

4.**Assemble the shepherd's pie:**

* Transfer the filling to a 9x13-inch baking dish.
* Spread the mashed potato topping evenly over the filling, making sure to cover the entire surface.

5.**Bake the shepherd's pie:**

* Place the dish in the preheated oven and bake for 25-30 minutes, or until golden and bubbling.

