# OpenReview forum: "Efficient Training-Free Multi-Token Prediction via Embedding-Space Probing"
_ICML.cc/2026/Conference — ICML 2026 regular_

### Official Review · Reviewer_RSkA · 2026-03-11

**Soundness:** 3
**Presentation:** 2
**Significance:** 2
**Originality:** 3
**Overall Recommendation:** 4
**Confidence:** 2

**Summary:**

This paper proposes a training-free multi-token prediction method that accelerates LLM decoding by probing the base model with mask tokens constructed in the embedding space. These mask tokens enable the model to generate speculative future tokens in parallel, which are organized into a dynamic token tree and verified by the base model to ensure lossless generation. Experiments demonstrate improvements in block efficiency and token throughput compared with existing training-free methods.

**Compliance With Llm Reviewing Policy:**

Affirmed.

**Final Justification:**

My main concerns have been addressed, and I have adjusted my score accordingly. The authors should clarify the notation and include the additional experimental results in the revision.

**Key Questions For Authors:**

See weaknesses.

**Limitations:**

yes

**Strengths And Weaknesses:**

Strengths：
1. The proposed method obtains multi-token predictions by probing the base model with mask tokens. It does not require additional training or auxiliary models, making it easy to apply to general LLMs without modifying the model architecture.
2. The experimental results on block efficiency and tokens per second demonstrate that the proposed method can effectively improve the efficiency of multi-token prediction.

Weaknesses：
1. The presentation is not good and contains confusing notation. For example, Figures 1 and 3 use many superscripts and subscripts, but their meanings are not clearly explained. In addition, Eq. (4) introduces the notation $m_i[s]$, which can be confused with the embedding notation $E[x_i]$ mentioned earlier. I strongly recommend that the authors clearly explain the meaning of each notation, for example by adding a notation table. Furthermore, Figure 5 on page 6 contains a large amount of empty space around it. The layout should be adjusted to improve the presentation.
2. The experiments only compare with training-free methods, while comparisons with methods that require training are missing. As a result, it is unclear whether the training-free advantage outweighs performance gaps.
3. The paper lacks analysis and discussion of failure cases, such as when mask probing fails or when the acceptance rate collapses. Providing such analysis would help offer a more comprehensive understanding of the proposed method.

---

> ### Author Rebuttal · Authors · 2026-03-31
>
> We thank the reviewer for the feedback and for recognizing the strengths of the proposed training‑free multi‑token prediction framework. We address each concern below.
>
> >W1. Confusing notation and presentation
>
> We agree and will revise the paper to make the notation in the Background (Sec. 2), Fig. 1, and the Method (Sec. 3) unambiguous. Concretely, we will:
> - Add a notation table defining all superscripts, subscripts, and symbols.
> - Clearly distinguish indices for token positions, tree nodes, and mask slots.
> - Rename overloaded symbols where needed (e.g., embeddings vs. indexed tokens).
>
> We will standardize notation as follows:
> - Target (accepted) tokens embeddings: $x_i$​
> - Future/draft token embeddings: $\hat{x}_i$
> - Mask tokens: $m_i^{j}$​, the $j$-th mask attached at position $i$ (e.g., $m_i^{1}$, $m_i^{2}$
> When relevant, we will explicitly state whether masks are attached to $x_i$​ or $\hat{x}_i$​. We will also add cross‑references and include the notation table for quick lookup.
>
>
> >W2. Missing comparison with training‑based methods
>
> We report speedup ratios relative to non‑speculative (autoregressive) decoding for training‑free baselines and EAGLE3. Among training‑free methods, our approach achieves the strongest speedups, with gains becoming more pronounced at larger model sizes (e.g., Qwen3‑32B). All methods within each table are measured under the same runtime setup (H100s here; the paper reports A100s).
> For training‑based speculative decoding, we compare against EAGLE3:
> - LLaMA‑3.1: We will ran EAGLE3 results using official draft checkpoints, setting it up with SpecBench
> - Qwen3‑32B: No official EAGLE3 checkpoint exists; obtaining results would require distillation data generation and multi‑GPU training which is out of our scope.
>
> | Model                  | BC |  PLD | STAND | LADE | Ours | GPUh | EAGLE3 | E3 GPUh            |
> |------------------------|----|------|-------|------|------|-----:|--------|--------------------|
> | Llama-3.1-8B-Instruct  | 30 | 1.20 | 1.13  | 1.10 | 1.37 | 0    | 2.36   | ~200–300           |
> | Llama-3.1-8B-Instruct  | 60 | 1.20 | 1.13  | 1.21 | 1.34 | 0    | 2.61   | ~200–300           |
> | Qwen3-32B              | 30 | 1.22 | 1.19  | 1.19 | 1.49 | 0    | N/A    | ≫ 200–300         |
> | Qwen3-32B              | 60 | 1.22 | 1.21  | 1.26 | 1.54 | 0    | N/A    | ≫ 200–300         |
>
> In summary, while EAGLE3 can achieve higher speedups, it is training‑based and incurs auxiliary‑model training and deployment overhead. Our work targets a complementary point: a single‑model, training‑free approach that exploits the target model’s latent ability to propose limited but reliable lookahead.
>
> >W3. Lack of failure case analysis
>
> We do not observe collapse in acceptance behavior, but agree it is important to discuss limitations. First, in some downstream settings (e.g., RAG), other training‑free baselines can be competitive or slightly better (e.g., STAND in Fig. 4). Second, we observe the lowest acceptance rates on ROLEPLAY, which is highly open‑ended and likely induces higher next‑token entropy. We will expand the Appendix with a brief failure‑case analysis, including representative examples and acceptance traces.
>
> Appendix G.7 already provides qualitative examples showing that acceptance is not limited to trivial continuations: the method sometimes accepts non‑trivial content tokens while rejecting plausible greedy ones. We will expand this analysis and view building targeted evaluations for such patterns as future work.
>
> >Closing remark
>
> We believe these clarifications strengthen the paper’s clarity, positioning, and completeness, while preserving the core contribution: a simple, training‑free mechanism for extracting latent future‑token signal from pretrained LLMs.

---

> > ### Author Rebuttal · Reviewer_RSkA · 2026-04-04
> >
> > Thank you for your response, which largely addresses my concerns. The paper should clarify the notation in the revision.

---

> > > ### Author Response · Authors · 2026-04-04
> > >
> > > We thank the reviewer for acknowledging our rebuttal and for finding our responses detailed and satisfactory. We also note that the reviewer selected the option “(a) Fully resolved – My concerns have been adequately addressed.”
> > >
> > > We are glad that all previously raised concerns have been addressed, including the comparison table of speed ratios with non‑autoregressive and training‑based setups, clarification of notation, and the discussion of failure cases.
> > >
> > > We fully respect the reviewer’s discretion in their evaluation. We simply wanted to ask whether maintaining the original score (3) after noting that all concerns were fully resolved was intentional, or if the reviewer would like to revisit the overall assessment in light of the clarifications provided.

---

### Official Review · Reviewer_7FLG · 2026-03-13

**Soundness:** 3
**Presentation:** 2
**Significance:** 2
**Originality:** 3
**Overall Recommendation:** 3
**Confidence:** 2

**Summary:**

This paper proposes a training-free MTP method by probing the model’s embedding space using dynamically generated mask tokens. It enables lossless generation without modifying model weights or training auxiliary draft models.

**Compliance With Llm Reviewing Policy:**

Affirmed.

**Final Justification:**

This is a borderline paper. The authors have addressed my concerns in the rebuttal. However, evidence shows that this paper is not well prepared: the paper layout is bad; the experiment setting is not aligned (A100 in the paper and H100 in the rebuttal). So I will keep my current rating.

**Key Questions For Authors:**

1. What is the additional memory and compute overhead introduced by maintaining the speculative token tree and mask tokens?
2. How sensitive is performance to mask token initialization and update rules?

**Limitations:**

yes

**Strengths And Weaknesses:**

Strengths:
1. This training-free framework is appealing from a deployment perspective.
2. The dynamic tree construction improves over fixed tree shapes.
---
Weaknesses:
1. Speedup ratio over standard generation is missed. The improvements on BE are relatively modest. It is unclear whether these gains translate into significant practical speedups in real-world inference systems.
2. Figures and Tables are badly organized in page 6.
3. The analysis based on cosine similarity provides intuition but does not clearly explain why the proposed probing strategy leads to improved multi-token prediction in practice.

---

> ### Author Rebuttal · Authors · 2026-03-31
>
> We thank the reviewer for the clear and constructive feedback and for recognizing the appeal of a training‑free framework and the benefits of dynamic tree construction. We respond to each point below.
>
> >W1. Missing speedup ratio over standard generation
>
> We agree that including a non‑speculative (autoregressive) baseline is important. The table below reports speedup ratios relative to standard AR decoding for training‑free methods and shows that our approach achieves the strongest speedups among training‑free baselines. The relative gains become more pronounced at larger model sizes (e.g., Qwen3‑32B), suggesting that the benefit persists—and can increase—as the target model scales. These results were obtained on H100s, while the paper reports numbers on A100s.
> | Model | BC | PLD | STAND | LADE | Ours |
> |---|---:|---:|---:|---:|---:|
> | Llama-3.1-8B-Instruct | 30 | 1.20 | 1.13 | 1.10 | 1.37 |
> | Llama-3.1-8B-Instruct | 60 | 1.20 | 1.13 | 1.21 | 1.34 |
> | Qwen3-32B | 30 | 1.22 | 1.19 | 1.19 | 1.49 |
> | Qwen3-32B | 60 | 1.22 | 1.21 | 1.26 | 1.54 |
>
>
> >W2. Figures and tables organization
>
> We agree and will improve the organization and layout of figures and tables, including resizing and reordering them to better align with the narrative flow and improve readability.
>
> >W3. Cosine similarity analysis does not fully explain why probing works
>
> We agree that explaining why high cosine similarity emerges is a deep and non‑trivial question. Our goal here is narrower: we empirically observe that accepted future tokens coincide with high cosine similarity and formalize why this signal leads to token acceptance under speculative decoding. While intuitive, a formal connection between cosine similarity and acceptance was missing; Lemma 1 provides this link.
> Lemma 1 establishes a connection between (i) representation‑level alignment (high cosine similarity between mask embeddings and next‑token states) and (ii) algorithmic success (the correct future token appearing in the top‑K draft set and being accepted). This implication is not obvious a priori. Lemma 1 makes it explicit, and Figure 2 empirically shows that higher cosine similarity correlates with higher acceptance rates. We will clarify that Lemma 1 does not explain the origin of the similarity, but justifies why embedding‑space probing yields valid speculative proposals when such similarity is present.
>
> >Q1. Additional memory and compute overhead
>
> Compute overhead is controlled by Block Complexity (BC), which is held fixed across all compared methods; thus all methods incur the same additional compute budget by design.
> Regarding memory: (i) mask tokens require storing only hidden‑dimension × number‑of‑masks, which is negligible relative to model parameters and KV cache; and (ii) no auxiliary models or additional decoder layers are introduced. We can add a brief quantitative discussion in the appendix if needed.
>
> >Q2. Sensitivity to mask token initialization and update rules
>
> Table 5 shows that the method is robust to mask initialization. In Table 12 (Appendix), we further show that initializing the mask embedding outside the effective support of the embedding space reduces performance, with average accepted tokens dropping by ≈2.5%.
> We also ablate the mask update coefficient $\lambda$. The paper uses $\lambda$=0.1; we additionally test $\lambda \in \{0.01, 0.5\}$ on Llama3.2‑3B and Llama3.1‑8B over SpecBench tasks. Large $\lambda=0.5$ updates too aggressively and reduces acceptance, while very small $\lambda=0.01$ under‑reacts to new tokens. Overall, $\lambda=0.1$ provides the best trade‑off.
> | Model | BC | λ=0.1 | λ=0.01 | λ=0.5 |
> |---|---:|---:|---:|---:|
> | Llama3.2-3B-Instruct | 30 | 1.59 | 1.53 | 1.43 |
> | Llama3.2-3B-Instruct | 60 | 1.67 | 1.63 | 1.52 |
> | Llama3.1-8B-Instruct | 30 | 1.62 | 1.60 | 1.47 |
> | Llama3.1-8B-Instruct | 60 | 1.71 | 1.69 | 1.57 |
>
>
> >Closing remark
>
> We hope these additional results and clarifications address the reviewer’s concerns and improve clarity. In particular, the new comparison against standard AR decoding demonstrates meaningful acceleration in the training‑free setting.

---

> > ### Author Rebuttal · Reviewer_7FLG · 2026-04-04
> >
> > Appreciate the response, which addresses my concerns.

---

> > > ### Author Response · Authors · 2026-04-04
> > >
> > > We thank the reviewer for acknowledging our rebuttal and for finding our responses detailed and satisfactory. We also note that the reviewer selected the option “(a) Fully resolved – My concerns have been adequately addressed.”
> > >
> > > We are glad that all previously raised concerns have been addressed, including the comparison table of speed ratios with non‑autoregressive baseline, clarification on memory overhead, and ablation on sensitivity to update rule.
> > >
> > > We fully respect the reviewer’s discretion in their evaluation. We simply wanted to ask whether maintaining the original score (3) after noting that all concerns were fully resolved was intentional, or if the reviewer would like to revisit the overall assessment in light of the clarifications provided.

---

### Official Review · Reviewer_Ai4m · 2026-03-13

**Soundness:** 3
**Presentation:** 2
**Significance:** 2
**Originality:** 3
**Overall Recommendation:** 4
**Confidence:** 3

**Summary:**

The paper introduces a speculative decoding method that doesn't require any auxiliary model. To achieve this, the authors insert mask tokens at the end of the prompt, argue that the model *pushes* these masking tokens' embeddings towards valid tokens, and use the output logits at those positions as the speculated next tokens. This introduces a *tree* of possible continuations (applying top-k on each of the masked tokens). After pruning the tree using a few ideas, all the remaining nodes of the tree are verified/rejected through a massive forward pass (with a careful attention mask). Notably, the verification happens at the same time as injecting the next mask tokens after each node and speculating the next tree. The extra cost introduced by the method is controlled by a block complexity parameter. Experiments show improvement over other model-free baselines.

**Compliance With Llm Reviewing Policy:**

Affirmed.

**Final Justification:**

I find the authors' rebuttal fully convincing, hence increasing my score. However, I'm not recommending "Accept" due to some clarity issues in the text and the limitation that auxiliary-model-free methods are going to inevitably be worse that auxiliary-model methods.

**Key Questions For Authors:**

1. Can the authors please add the non-speculative baseline to Table 1?
2. The method is described to support any depth, but in practice, the authors only go as far as depth 2, which basically means the speedup the method can provide is severely limited. Can the authors clarify what happens at larger depths? Is it simply a memory problem or the mask tokens predictions turn out to be invalid at that depth?
3. Can the authors clarify the tree expansion method? Do you only expand the top token (as the text suggests) or every token (as the algorithm suggests)?
4. The choice of mask token (the prompt average), although working better than other compared options in practice, still sounds arbitrary with no theoretical or intuitive explanation of why this is a good idea. I would be interested to know if the authors have any deeper understanding of this. Also, the e2e results are somewhat similar across different mask tokens (Table 5), hence I would suspect Figures similar to Fig 2 could be generated for other masking tokens as well, hence I don't think Fig 2 addresses the selection of this specific masking token.

In general, while I believe this is an interesting paper, I believe there are gaps that need to be addressed/resolved. I would be open to increasing my score depending on how the rebuttal goes.

**Limitations:**

While not specifically clear from the paper, I would suspect the gains of auxiliary-model-free methods such as this paper are expected to be much less compared to that of methods with auxiliary models. I think discussing this point in the paper would be helpful in positioning the paper in the literature.

**Strengths And Weaknesses:**

Strengths:
1. Generally interesting and well-thought-out algorithm.
2. Extensive experiments, including lots of ablation studies.
3. Fast implementation.

Weaknesses (see questions):
1. The theoretical results (Lemma 3.1) are superficial. The authors prove: if similarity is high  => the right answer is in top-k, which I would argue is somewhat trivial. While what we're interested in is that why is the similarity high. Not sure how one would approach this for an LLM, but the provided theory doesn't provide any non-trivial insight.
2. The method is generally explained poorly in my opinion (though understandable with some effort). It would be helpful to (1) explain exactly what the Top-K functions in sections 3.3 and 3.4 are applied to. It seems that top-k sometimes is applied on the tree nodes and sometimes to the logits. It would be helpful to be clear about this. (2) Also, including an example (with real understandable text) would be helpful.  (3) the algorithm does not match the text. From the text, it seems that only top-1 (highest probability) token is allowed to expand, while in the algorithms, all tokens expand (line 7). Generally, I believe the writing of the paper could be improved.
3. Token-per-second is not reported for the default single-token (non-speculative) decoding baseline, which makes understanding the actual gain of the method difficult. Given that speculative methods that use auxiliary methods typically offer over 3x speedup [1], it would be interesting to see where this method stands.

[1] https://arxiv.org/pdf/2406.16858

---

> ### Author Rebuttal · Authors · 2026-03-31
>
> We thank the reviewer for the careful reading, for appreciating the algorithmic design and extensive ablations, and for the constructive suggestions. Below we address each concern in detail.
>
> >W1. Theoretical results appear superficial
>
> We agree that explaining why high cosine similarity emerges is a deep and non‑trivial question. Our goal here is narrower: we empirically observe that accepted future tokens coincide with high cosine similarity and formalize why this signal leads to token acceptance under speculative decoding. While intuitive, a formal connection between similarity and acceptance was missing; Lemma 1 provides this link.
> Specifically, Lemma 1 connects (i) representation‑level alignment (high cosine similarity between mask embeddings and next‑token states) with (ii) algorithmic success (the correct future token appearing in the top‑K draft set and being accepted). This implication is not obvious a priori. Lemma 1 makes it explicit, and Figure 2 shows that higher cosine similarity correlates with higher acceptance rates. We will clarify that Lemma 1 does not explain the origin of the similarity, but justifies why embedding‑space probing yields valid speculative proposals when such similarity is present.
>
> >W2(1). Ambiguity of Top K usage
>
> We agree and will revise the text to distinguish two uses of Top‑K:
>  - Top‑K over logits (per mask position): At each masked position, we take the Top‑K tokens from the output distribution. With multiple masks, we allow different $K_i$ per position. A draft tree is constructed from these candidates, retaining high‑probability trajectories while enforcing the block‑complexity constraint (Appendix D).
>  - After draft tree construction we attach mask tokens (Fig. 3) and run the base model for parallel verification and generation.
>
> >W2(2). Need for a concrete example
>
> We will add a short walkthrough and a compact text‑based schematic with text values for ease of understanding
>
> >W2(3). expanding top 1 token vs. all tokens
>
> Thank you for flagging this. We maintain a set of high‑probability draft trajectories, but at each expansion step branch only on the top‑1 next token (as in [1]). This avoids exponential tree growth and simplifies pruning; the procedure is illustrated in Appendix D.
> [1] Lin et al., BiTA, ESA 2025.
>
> >Q3. Missing non speculative baseline and comparison to EAGLE3 style methods
>
> Table below reports speedups vs. standard AR decoding for training‑free methods and EAGLE3. Our approach achieves the strongest speedups among training‑free baselines, with gains increasing at larger model sizes (e.g., Qwen3‑32B). We used H100s here; paper results use A100s.
>
> Regarding Eagle3:
>  - For LLaMA‑3.1, we ran EAGLE3 using official draft checkpoints and SpecBench.
>  - For Qwen3‑32B, no official EAGLE3 checkpoint exists; obtaining results would require distillation dataset generation and multi‑GPU training.
> ```
> | Model        | BC | PLD | STD | LADE | Ours | GPUh | EAGLE3 | E3 GPUh |
> |--------------|----|-----|-----|------|------|-----:|--------|---------|
> | Llama-3.1-8B | 30 | 1.20| 1.13| 1.10 | 1.37 | 0    | 2.36   | ~200-300|
> | Llama-3.1-8B | 60 | 1.20| 1.13| 1.21 | 1.34 | 0    | 2.61   |         |
> | Qwen3-32B    | 30 | 1.22| 1.19| 1.19 | 1.49 | 0    | N/A    | >200-300|
> | Qwen3-32B    | 60 | 1.22| 1.21| 1.26 | 1.54 | 0    |        |         |
> ```
> We will clarify that our method is a single‑model, training‑free approach, complementary to auxiliary‑model methods that can achieve larger speedups but require extra training and deployment complexity.
>
> >Q2. Limited depth (mostly depth 2) lookahead
>
> Even shallow lookahead yields non‑trivial gains in the training‑free setting: our method reliably generates 2 tokens per iteration (occasionally 3) without retraining the base model. As the model is autoregressive, increasing the number of masks under a fixed block‑complexity budget leads to diminishing returns.
> With BC = 120 and 3 mask tokens (4‑token lookahead), the acceptance distribution is [0.432, 0.468, 0.093, 0.007]. We will clarify that deeper lookahead is fundamentally harder for autoregressive LMs; our contribution is extracting the maximum reliable future signal (≈ 1–2 tokens) in a training‑free manner.
>
> >Q4. Mask token design and justification
>
> We do not claim the prompt‑average mask embedding is optimal, but use it as a stable default. We clarify that (i) it acts as a semantic prior keeping masks in‑distribution, (ii) we perform extensive ablations over initialization and update rules, and (iii) performance is robust across reasonable designs.
> We point to:
>  - Table 12: moving the mask embedding farther from the embedding‑space mean reduces accepted tokens by ~2.5%.
>  - Yes Fig.2 is not to support our design rather to show correlation between cosine similarity and acceptance; Lemma 1 proves that, conditional on high similarity, the correct future token appears in the Top‑K draft set and can be accepted.
>
> >Limitations
>
> We'll add trade-off with trained drafts in Appendix

---

> > ### Author Rebuttal · Reviewer_Ai4m · 2026-04-03
> >
> > I would like to thank the authors for the clarifications. I am fully convinced by their rebuttal, and will accordingly increase my score to "Weak Accept." I would encourage the authors to incorporate the discussions from this rebuttal in the next revision of the paper, especially the ones regarding clarity of the method and the comparison with auxiliary-model baselines.

---

> > > ### Author Response · Authors · 2026-04-03
> > >
> > > We thank the reviewer for their thoughtful follow‑up and are glad that our rebuttal addressed their questions. We will incorporate the discussed clarifications and changes in the final version of the paper, and we appreciate the reviewer’s updated evaluation.

---

### Official Review · Reviewer_oaCx · 2026-03-16

**Soundness:** 2
**Presentation:** 2
**Significance:** 3
**Originality:** 2
**Overall Recommendation:** 3
**Confidence:** 2

**Summary:**

The authors speed up text generation in Large Language Models by introducing a training-free multi-token prediction method. The method works by producing stand-in mask token representations for the unseen future tokens, which allow the model to draft multiple tokens ahead. The authors use speculative decoding (both with temperature 0 and 1) to guarantee that the generation quality of the original LLM is retained and introduce bespoke draft trees to further speed up generation. The method is compared to other training-free methods (PLD, STAND, LADE) for the LLaMA3 and Qwen3 models on a subset of the SpecBench tasks. The computational budget of all methods is constrained according to the block complexity metric introduced, to put competing models on an equal footing. Under these constraints, the introduced method increases throughput by up to 15-19%.

**Compliance With Llm Reviewing Policy:**

Affirmed.

**Final Justification:**

The authors provided detailed and thoughtful responses that have addressed all of my concerns. While I considered raising my score, I cannot do so with confidence because I am not enthusiastic enough about the theoretical contributions/insights and because I believe the writing needs to be made more precise.

**Key Questions For Authors:**

1. Introducing a metric like block complexity makes sense, since it makes the comparison between models precise. However, it is unclear what reasonable values for block complexity are. Could the authors explain what the motivation for choosing the values of 10, 30 and 60 is? How do methods compare if the block complexity is increased?
2. On line 206, the authors mention: "during prefill, we expand only the top-1 token at each depth". I believe I must have misunderstood something, could the authors please elaborate on why they expand tokens during prefilling given that all activations can be computed in a single forward pass?
3. Could the authors please further clarify what they mean by temperature on line 226 (right)?
4. Could the authors elaborate on their motivation for introducing the Block Efficiency metric and naming it something different from the more commonly used jargon such as mean acceptance/generation length?

**Limitations:**

It would be useful if the authors could elaborate on any limitations of their method in terms of a) how far into the future it can predict and b) how the method scales if we increase block complexity.

**Strengths And Weaknesses:**

## Strengths

1. (**Significance**) Speeding up LLM generation while maintaining its generation quality is an important problem. Moreover, being able to do so in a training-free way is alluring and impactful.
2. (**Soundness**) The authors conduct many experiments on standard benchmarks and compare against reasonable baselines, showing promising improvements over those baselines.
3. (**Significance**) The question of how much signal for MTP is present in the embedding space of LLMs and can be extracted via probing is interesting.

## Weaknesses

1. (**Presentation / Soundness**) The acceptance criteria and algorithms used in the paper to guarantee lossless generation have not been specified. In Equation 3, the authors only describe the greedy decoding case, i.e. temperature 0, and the citation (Lin 2024) also only discusses greedy decoding with a draft tree. However, in the paper the authors also discuss sampling with temperature 1. Since the authors claim their method is lossless, it is important to know what theoretical result this guarantee stems from. Can the authors clearly write which acceptance criterion they are using in each case and how they retain their guarantees when using the draft tree? The wording later on in the paper is also unclear. For example, on line 214, the authors mention: "Each predicted token is verified by comparing it with the base model's next token distribution, and is accepted only if it matches exactly, ensuring lossless generation". But it is unclear what is meant, because it is unclear how a predicted token can be compared against a distribution. Lastly, on line 270 the authors say: "We use sample matching, enabling lossless generation. where a token is accepted only if it is an exact match, enabling lossless generation". As with earlier references, this also seems to be referring to the temperature 0 case only, but 10 lines later both temperature 0 and temperature 1 are mentioned. It would useful if the authors could clarify what they mean by temperature in this context. Moreover, they should specify the algorithms they are using and cite the related work they are building on in a clear, unambiguous way.
2. (**Soundness/Originality**) Some claims in the paper seem unsubstantiated or exaggerated beyond what the evidence and content in the paper suggests. a) The authors claim that "LLMs possess latent multi-token prediction abilities" and that they introduce a novel MTP paradigm that enables multi-token generation. However, it is a bit unclear how far into the future their probing method can predict given that 1 + mean acceptance length is less than 2 for all experiments - and this is confirmed by the key takeaway on line 340 where they mention: "LLMs, when probed appropriately, can confidently predict an additional token with minimal overhead". It would be great if the authors could report a histogram over the number of accepted tokens to back up their MTP claims, or alternatively, relax their claims to be about predicting an additional token and not multiple tokens in general. b) The authors claim: "adding additional draft decoder layers" makes other methods unsuitable for edge devices. However, there is not enough evidence or information provided to understand why this is the case. Moreover, what makes their method suitable for edge devices? They mention that low BC is important, but it would be great if they could provide more context: why is an expensive speculative decoding framework the constraint and not the size of the models themselves, for instance?
3. (**Presentation**) While the basic idea was straightforward to follow, the narrative gets complicated by details that seem not to be central to the main story. The presentation could be improved by better sign-posting and motivating additional details when they are being introduced. For example, a) three alternative strategies are proposed for constructing embeddings for the masked tokens, but the reader is not told why 3 alternatives are needed. Moreover, in the main experiments, it seems that only strategy c, the embedding mean, is used. Perhaps it would make sense to only introduce the additional embedding methods when the ablation is discussed in section 4.4. b) The discussion about introducing two embeddings per mask token on line 064 (right) is confusing, because it appears suddenly with no motivation. What is expected to be the benefir of having 2 embeddings per mask token? (also this only seems to appear for block complexity 60) c) Figure 5 is too small to read and (nitpick) Figure 4 seems to not print well on paper - the patterned columns did not show up at all. d) The use of the term next-next is a bit confusing, initially it seems like a typo, but presumably the authors mean token_{t+2}. It would be helpful if they explicitly introduce notation for this. e) In Figure 2, what masking strategy is the result for?
4. (**Soundness**) In Lemma 1, the authors show that if the mask embedding they produce for the token at t+1 and the embedding that would have been produced for next token prediction at t+1 have large enough cosine similarity, then the argmax for token at t+2 will be among the top-k tokens produced by the mask embedding for that token. While this lemma is nice to have, it is unclear how it advances the story of the paper, since the authors do not have any guarantees that their mask embeddings have high cosine similarity with the next token embedding. Can the authors please elaborate on how Lemma 1 supports their approach and how it motivates their choice of mask embedding?




### Typos

* Line 20 (right): Li et al 2024 should be \citet and not \citep.
* Line 136: Probably need to specify a constraint such that the index $t-k-i$ cannot be $\leq 0$?
* Line 139 : be standard deviation -> be the standard deviation
* Line 147: Index i is used twice (both to range 1...t and 1...k)
* Line 253: use single mask -> use a single mask
* Line 255: use of " quotes instead of TeX quotes
* Line 259: average accepted tokens -> average generated tokens
* Line 270: sample matching. -> sample matching,
* Line 413: Our theoretical insight reasons why -> Our theoretical insight explains why
* Figure 1 caption: verfication -> verification

---

> ### Author Rebuttal · Authors · 2026-03-31
>
> We thank the reviewer for the detailed feedback!
>
> >W1. Acceptance criteria, lossless generation (T=0, T=1)
>
> We use sample matching as the acceptance rule for both greedy decoding (T=0) and temperature‑1 sampling (T=1). A draft token is accepted only if it exactly matches the token the base model would produce at that position under the same decoding procedure, as in prior speculative decoding systems including EAGLE3 [1].
> For T=0, the draft token must match the base model’s greedy output; for T=1, it must match the token sampled from the base model using the same temperature and randomness source. While stricter than the original formulation [2], sample matching is commonly used in tree‑based drafting, where relaxed acceptance requires additional care (e.g., recursive speculative decoding [3]). Training‑based draft models such as EAGLE3 [1] also adopt this rule for T=1.
> Since all accepted tokens exactly match the base model’s output, the procedure is lossless by construction. We will revise the paper to explicitly state this acceptance criterion for both T=0 and T=1.
>
> >W2a. Scope of multi‑token prediction and acceptance statistics
>
> We agree that the most consistent gains come from predicting one additional token ahead and will tighten the language to avoid over‑general claims. Table 10 (Appendix) compares predicting 2 vs. 3 tokens ahead (1 vs. 2 mask positions) at BC=60. While mean accepted length is similar, we observe a consistent trend: open‑ended generation benefits more from 2‑token lookahead with larger top‑K expansion, whereas closed‑ended generation benefits more from 3‑token lookahead with smaller top‑K expansion, where added structure makes shallow multi‑step lookahead feasible.
> We ran at BC=120 with 3 mask positions and top‑K [10,10,9] (4‑token lookahead), marginal gains diminish; the accepted‑token distribution is [0.432, 0.468, 0.093, 0.007]. Overall, increasing lookahead yields diminishing returns, though the model reliably supports at least a 2‑token lookahead.
>
> >W2b. Suitability for edge settings and draft‑model overhead
>
> We do not claim draft‑model‑based methods are infeasible on edge devices. Rather, when the base model is fixed, acceleration methods should avoid additional parameters, memory residency, or serving complexity.
> Our method is attractive in deployment‑constrained settings because it is training‑free, requires no auxiliary model, and uses only the base model at inference time. In contrast, training‑based draft methods typically require dataset generation and retraining when switching base models (see [4]). Our approach introduces no additional learned parameters and avoids per‑model retraining, which is especially relevant in memory‑constrained or deployment‑sensitive environments.
>
> >W3. Presentation issues
>
> We will defer alternative mask embeddings to ablations; clarify the motivation and usage of two mask tokens; improve figure readability (Figures 4 and 5); replace informal terminology with explicit notation and add a notation table; and explicitly state the masking strategy used in Figure 2.
>
> >W4. Role of Lemma 1 and cosine similarity analysis
>
> Explaining why high cosine similarity emerges or what is the optimal design of mask tokens are non-trivial questions. Table 5 shows the robustness of mask tokens to initializations. Additionally, using mean of prompt design we empirically observed that accepted tokens have high cosine similarity (Fig. 2). Lemma 1 is a formal proof as to how representation‑level signal leads to acceptance under speculative decoding. This implication is not obvious a priori.
>
> Q1.
> We will clarify that BC=10/30/60 correspond to low/medium/high incremental compute regimes commonly used in prior work, and add BC=120 results to show that higher BC can reduce throughput due to GPU throttling and memory pressure even though it gives higher accepted tokens.
> | BC | 30 | 60 | 120 |
> |----|----|----|-----|
> | Avg tokens accepted | 1.62 | 1.72 | 1.78 |
> | Token rate | 43.68 | 42.82 | 40.78 |
>
> Q2.
> “Expansion” refers to speculative tree construction from logits, not additional Transformer forward passes during prefill. During prefill mask tokens are appended at the end of prompt, the corresponding output logits are used to create draft tree.
>
> Q3.
> Addressed in W1
>
> Q4.
> We note that prior work also reports Block Efficiency [3–5].
>
> [1] Li, Y., et al. EAGLE‑3: Scaling inference acceleration of LLMs via training‑time tests. arXiv:2503.01840, 2025.
>
> [2] Leviathan, Y., Kalman, M., and Matias, Y. Fast inference from transformers via speculative decoding. ICML, 2023.
>
> [3] Jeon, W., et al. Recursive speculative decoding: Sampling without replacement for LLM inference. arXiv:2402.14160, 2024.
>
> [4] Goel, R., et al. Direct alignment of draft models for speculative decoding with chat‑tuned LLMs. arXiv:2403.00858, 2024.
>
> [5] Gagrani, M., et al. Speculative decoding for multimodal large language models. CVPR, 2024.

---

> > ### Author Rebuttal · Reviewer_oaCx · 2026-04-04
> >
> > I thank the authors for their detailed response and for resolving many of my concerns. I appreciate their honest acknowledgement about the maximum look-ahead their method can support while increasing throughput and the additional results showing that for 3 mask positions with BC=120 their method can accept more than one token (accepted token distributions [0.432, 0.468, 0.093, 0.007]). I also appreciate the changes proposed by the authors, though I have some reservations about whether these changes can be incorporated without a further round of review.
> >
> > I have a further question regarding the speculative decoding algorithm:
> >
> > W1. I am still confused whether their approach for T=1 is the same as what they cite in the literature, because a) The authors mention that "*A draft token is accepted only if it exactly matches the token the base model would produce at that position under the same decoding procedure, as in prior speculative decoding systems including EAGLE3 [1] ..(omitted).. for T=1, it must match the token sampled from the base model using the same temperature and randomness source.*". However, the EAGLE 3 paper in Section 2.1 claims to use Leviathan et al. where rejection sampling is carried out by comparing the target model's probabilities p to the draft model's probabilities q. Confusingly, the EAGLE 3 code instead samples candidates and compares p to a one-hot distribution by setting [qx=1](https://github.com/SafeAILab/EAGLE/blob/cb7e0841fe0c206c6ed74a197ad5e2a1f13f5a2b/eagle/model/utils.py#L398), see also this [github issue](https://github.com/SafeAILab/EAGLE/issues/329) that also highlights the discrepancy between what is stated in the EAGLE 3 paper and the code. It is unclear to me that either of these cases are equivalent to what the authors call sample matching. [In the EAGLE 3 code](https://github.com/SafeAILab/EAGLE/blob/cb7e0841fe0c206c6ed74a197ad5e2a1f13f5a2b/eagle/model/utils.py#L400), when T=1 the accepted tokens change depending on the proposed candidates, so the accepted tokens would not generally agree with the tokens generated from the base model at a specific temperature and randomness source. Could the authors please clarify if they are following the EAGLE 3 code or doing something different?
> > b) Has the term "sample matching" been introduced previously in the literature or is this a term the authors introduce?
> >
> > Q4. I thank the authors for the clarification, it would be useful if they cited these papers when introducing the metric.

---

> > > ### Author Response · Authors · 2026-04-04
> > >
> > > We thank the reviewer again for their feedback and hope for our paper
> > >
> > > Since our method is fully training-free, we also note that any clarifications and wording changes discussed below can be cleanly incorporated into the final version.
> > >
> > > >W1 Clarification
> > >
> > > Thank you for raising this—this point is indeed easy to confuse due to differences between how some papers describe speculative sampling and how certain implementations realize it in practice.
> > >
> > > - (a) What we mean by “sample matching” (T=1):
> > >
> > > In our implementation, what we refer to as sample matching corresponds to verifying draft tokens by matching them against the target model’s one-hot outcome at verification time. That is, a draft token is accepted if and only if it exactly matches the token selected by the target model under its own decoding rule (greedy or sampling).
> > > This behavior matches the implementation used in the EAGLE-3 GitHub code, where the verification step effectively treats the proposal distribution $q$ as a one-hot distribution on the proposed token, rather than performing full rejection sampling using draft probabilities. In other words, acceptance is based on exact token equality, not on comparing the full $p$ vs. $q$ distributions as described in the generic speculative sampling formulation (e.g., Leviathan et al.).
> > > Importantly, this implementation-level behavior differs from the abstract rejection-sampling description, but it is consistent with what is actually deployed in EAGLE-3-style systems.
> > >
> > > We also note that the draft token tree generation in EAGLE-3 is independent of the target model’s temperature ([ref code](https://github1s.com/SafeAILab/EAGLE/blob/main/eagle/model/cnets.py#L670)), as the draft tree is constructed before verification. As a result, at higher target temperatures, acceptance rates can decrease—since sampled target tokens may diverge from the draft proposals—even though the draft tree itself remains unchanged. This is where the tree-based structure becomes important, as it provides multiple candidate continuations that can still be matched under higher-temperature target sampling.
> > >
> > > - (b) Terminology: Sample Matching
> > >
> > > To the best of our knowledge, the phrase “sample matching” is not a standardized term used broadly in prior speculative decoding papers or mainstream repos.
> > >
> > > We use “sample matching” purely as a descriptive label for this strict one‑hot (exact‑equality) acceptance rule. Here, “matching under the same decoding procedure” means that the target uses the same decoding rule at verification time, but the accepted tokens still depend on which tokens the draft proposes.
> > >
> > > We agree that this distinction was not sufficiently explicit in the original text, and we will reword this description in the final version of the paper to avoid ambiguity.
> > >
> > > Finally, we emphasize that under this exact acceptance criterion, our method still consistently outperforms other training-free baselines, as shown in Tables 7 and 8 in the Appendix.
> > >
> > > >Q4
> > >
> > > Sure we will cite these paper
> > >
> > > We hope that this addresses all of the reviewer’s concerns.

---

### Decision · Program_Chairs · 2026-04-30

**Decision:**

Accept (regular)

**Comment:**

The rebuttal addressed several key reviewer concerns, particularly regarding method clarity, empirical justification, and comparison to related baselines. Reviewers acknowledged these improvements, and the overall discussion suggests the contribution is meaningful and technically sound. While some concerns remain about presentation and scope, they do not outweigh the paper’s novelty and practical value.